



# Geochemical and microstructural characterisation of two species of cool-water bivalves (*Fulvia tenuicostata* and *Soletellina biradiata*) from Western Australia.

Liza M. Roger[1,2], Annette D. George[1], Jeremy Shaw[2], Robert D. Hart[2], Malcolm Roberts[2], Thomas
5   Becker[2,3], Bradley J. McDonald[4] and Noreen J. Evans[4].

[1]School of Earth Sciences, The University of Western Australia, Crawley, 6009, Australia
[2]Centre for Microscopy, Characterisation and Analysis, The University of Western Australia, Crawley, 6009, Australia
[3]Department of Chemistry, Nanochemistry Research Institute, Curtin University, GPO Box U1987, Perth, 6845, Australia
[4]Department of Applied Geology, John de Laeter Centre, TIGeR, Curtin University, Bentley, 6102, Australia

10   *Correspondence to*: Liza M. Roger (liza.roger@hotmail.fr)




**Abstract.** The shells of two marine bivalve species (*Fulvia tenuicostata* and S*oletellina biradiata*), endemic to south Western Australia, have been characterised using a combined crystallographic, spectroscopic and geochemical approach. Both species have been described previously as purely aragonitic, however, this study identified the presence of three phases, namely aragonite, calcite and Mg-calcite using XRD analysis. Data obtained via confocal Raman spectroscopy, electron

5  probe microanalysis, and laser ablation inductively coupled plasma - mass spectrometry (LA ICP-MS) show correlations between Mg/S and Mg/P in *F. tenuicostata*, and Sr/S and S/Ba in *S. biradiata*. The composition of organic macromolecules that constitute the shell organic matrix (i.e. soluble phosphorus-dominated and/or insoluble sulphur-dominated fraction) influences the incorporation of Mg, Sr and Ba into the crystal lattice. Ionic substitution, particularly $Ca^{2+}$ by $Mg^{2+}$ in calcite in *F. tenuicostata*, appears to have been promoted by the combination of both S- and P-dominated organic macromolecules.

10  The elemental composition of these two marine bivalve shells is species-specific and is influenced by many factors such as crystallographic structure, organic macromolecule composition and environmental setting. In order to reliably use bivalve shells as proxies for paleoenvironmental reconstructions, both the organic and inorganic crystalline material need to be characterised to account for all influencing factors and accurately describe the "vital effect".

15  Keywords

Geochemistry, microstructure, bivalve, organic macromolecule, crystal lattice, Mg/Ca, Sr/Ca, aragonite, calcite





## 1 Introduction

As calcifiers, molluscs play an important role in the ocean carbonate cycle. The calcium carbonate formed by marine organisms is a complex geochemical source and sink of carbon, which controls total oceanic carbon content, $p$CO$_2$ (partial pressure of CO$_2$) and more generally contributes to ocean alkalinity (e.g. Gattuso and Hansson, 2011). The biomineralisation

process, or more precisely the elemental and isotopic composition of biogenic CaCO$_3$, is affected by physiological processes (e.g. Freitas et al., 2006; Gillikin et al., 2005b; Lowenstam and Weiner, 1989) and environmental conditions at the time of deposition (e.g. Ferguson et al., 2011; Gazeau et al., 2010; Hahn et al., 2014; Heinemann, 2011; Henkes et al., 2013; Lowenstam and Weiner, 1989; Schöne et al., 2011). Molluscs are globally distributed and demonstrate sequential growth, thus providing high-resolution seasonal and sub-seasonal records of environmental conditions (e.g. Jones and Quitmyer,

10    1996).

A number of bivalve species are extremely long-lived, with lifetimes of many decades or even centuries, e.g. fresh water pearl mussels (e.g. Schöne et al., 2004), geoduck clams (e.g. Strom et al., 2004), ocean quahogs (e.g. Schöne et al., 2005) and giant clams (e.g. Ayling et al., 2006; Bonham, 1965). Many oceanic conditions, such as seawater surface temperature, productivity, circulation and carbon reservoir dynamics, have been reconstructed successfully using bivalve

shell records (e.g. Klein et al., 1996; Lazareth et al., 2003; Richardson et al., 2004; Wanamaker et al., 2011). However, an organism's physiological control on the chemical composition of its shell can affect the reliability of the parameters recorded (e.g. Gillikin et al., 2005b; Lazareth et al., 2013; Lazareth et al., 2003). Differences between measured isotopic and elemental composition in carbonates materials are typically attributed to a physiological/biological influence commonly called the 'vital effect' (Zeebe et al., 2008). Spero et al. (1991) first attempted to describe the vital effect through quantitative

modelling of stable carbon isotopes in foraminifera. The influence of the vital effect is variable between species, which is problematic for the development of a unique geochemical equilibrium model for seawater property reconstructions.

The work of pioneers in the field of molluscan biomineralisation, Bowerbank (Bowerbank, 1844) and Carpenter (Carpenter, 1845, 1847) amongst others, shifted the focus from fossil structures to 'modern' biogenic CaCO$_3$ structures (such as coral skeleton and mollusc shells). Influential work by Urey and colleagues (Urey et al., 1951) first used stable oxygen

isotope values ($\delta^{18}$O) in biogenic CaCO$_3$ (belemnite rostrum) as a proxy for seasonal temperature change. Certain limitations remained, e.g. separating $\delta^{18}$O$_{water}$ (which is linearly correlated to salinity) from the temperature signals, leading to the development of other proxies according to their temperature dependency, e.g. Sr/Ca and Mg/Ca ratios (Gillikin et al., 2005a; Shirai et al., 2014). However, further work showed that elemental composition is strongly controlled by biological factors including: growth rate (e.g. Gillikin et al., 2005c); calcification rate (e.g. Carré et al., 2006); ontogenetic age (e.g. Purton et

al., 1999); organic matrix (e.g. Schone et al., 2010; Takesue et al., 2008) and microstructure (overall fabrics) (e.g. Schöne et al., 2013; Shirai et al., 2014).

Long-term records of seawater parameters (e.g. temperature, alkalinity and pH) are essential for understanding past climatic and oceanic changes in the current context of global climate change. Data from diverse geographical locations are



needed to develop a global understanding of those changes. Records from bivalve shells have contributed considerably to our knowledge, but environmental controls need to be well defined and differentiated from biological and physiological factors (e.g. Jacob et al., 2008).

This study focuses on the microstructure and composition of shell from two species of bivalve endemic to
southwestern Australia to gain insight into the physio-chemical processes involved in molluscan shell calcification. It constitutes the first step towards monitoring global climate change through biogenic carbonate in southwestern Australia. In general, limited geochemical data are available for temperate Australian marine bivalves and, to our knowledge, this work is the first multimodal (crystallographic, spectroscopic and geochemical) study focusing on marine bivalve species from Australian waters. The structure and mineral composition of shells from the Western Australian bivalves, *F. tenuicostata* and
*S. biradiata,* have not previously been investigated as a potential proxy for future studies using this integrated approach. Given the absence of corals along the southern coast of Western Australia and the south coast of Australia as a whole, bivalve shells could serve as an alternative proxy for generating records of historical ocean chemistry providing that the overarching mechanisms are well defined and understood (e.g. elemental equilibrium, vital effect). This study contributes to the global understanding of how marine bivalves record environmental change through shell growth and the processes
governing marine biogenic carbonate production.

## 2 Materials and Methods

### 2.1 Study Area – Marine Setting

Samples were collected from King George Sound, between 3–5m depth, on the southern coast of Western Australia near the town of Albany (Fig.1). This area experiences seasonal freshwater inputs, a temperate to Mediterranean climate, mean
annual rainfall of $931.5 \pm 0.2$ mm between 1877 and 2014, maximum, ~$126.1 \pm 0.2$ mm, rainfall in August and minimum, ~$30.1 \pm 0.2$ mm, in December (Australia Government Bureau of Meteorology, Brouwers et al., 2013; Klausmeyer and Shaw, 2009). The main currents influencing the area are the Capes Current (coastal current) flowing westwards, the Leeuwin Current (shelf current) flowing eastward and the Flinders Current (oceanic current) flowing westward (Cresswell and Golding, 1980). Further south, Australian waters meet the cold and nutrient-rich Antarctic waters. The southwestern
coastline of Australia is impacted by strong winter storms influenced by pressure systems over the Great Australian Bight, as well as Leeuwin current dynamics. Seawater surface temperatures (SST) vary annually between 16°C–21°C (August/September to April/May) and bottom temperatures vary between 15°C–20.6°C (Australian Government Bureau of Meteorology), at 40–50 m maximum depth.



## 2.2 Species

The study uses two species of bivalves: *Fulvia tenuicostata* (Lamarck, 1819) ('thin-ribbed cockle') and *Soletellina biradiata* (Wood, 1815) ('double-rayed sunset clam', Fig. 2). Both species are native and endemic to southwestern Australia (WoRMS et al., 2015). *F. tenuicostata* (Mollusca : Bivalvia : Veneroida : Cardiidae), Fig. 2a and 2b, is typically 50-55 mm in length
(anterior to posterior margin). The shells of this species are cream-coloured becoming yellow towards the margin (the periostracum can show staining from the surrounding mud). The shells are thin and delicate with narrow ribs (Fig. 2a and 2b). This genus is found in moderately sheltered areas of muddy sand (WoRMS et al., 2015). The shell structure of *F. tenuicostata* consists of a prismatic outer layer (OL), a simple crossed-lamellar middle layer (ML) and a complex crossed-lamellar inner layer (IL). *S. biradiata* (Mollusca : Bivalvia : Veneroida : Psammobiidae) shells, Fig. 2c and 2d, may measure
up to 70 mm. Shells are thin walled and have an elongate shape. With the thick periostracum removed, the shells are cream-coloured with tinges of pink and purple, with two distinctive pale radial rays extending from the umbo to the growing edge. *S. biradiata* is also found in moderately sheltered areas with muddy sand substrates. The shell structure of *S. biradiata* consists of a prismatic OL composed of acicular prismatic crystals directed toward the outer surface and locally curved, a simple crossed-lamellar ML and a complex crossed-lamellar IL. The OL may be absent, in which case the lamellae of the
ML extend to the outer surface as noted for other species of this subfamily (Popov, 1986).

## 2.3 Sample Preparation

Shells (five *F. tenuicostata* and three *S. biradiata*) were rinsed with fresh water, manually cleaned of any organic residues by scrubbing, then soaked in sodium hypochlorite (12.3 vol%) for approximately 12 hours. Shells were thoroughly rinsed with Milli-Q water then air dried. According to Krause-Nehring et al. (2011) treatment of biogenic calcium carbonate to remove
organic matter may have an impact on elemental and crystallographic composition in the bivalve *Arctica islandica*. The effects of sodium hypochlorite (NaOCl) pretreatment (also used in our study) on *A. islandica* shell powder were insignificant compared to other treatments tested in that study. Indeed Sr/Ca and Mg/Ca ratios did not change significantly, and very little to no dissolution or phase change were observed. NaOCl treatment was also proved to be the most efficient to remove organic matter with the least effect on structure and composition in coral cores (Nategaal et al., 2012). Krause-Nehring et al.
(2012) performed their study on powdered, pre-cleaned (manual removal of the periostracum) and found very little effect. In our study, where intact shells were pretreated with NaOCl, we suggest that the geochemical and crystallographic composition of the bivalves studied here was not significantly altered despite having some hypochlorite solution percolate through.

Shells were sampled along the growing edge using a hand-held dental drill for x-ray diffraction analysis (Fig. 2). Fifty
milligrams of very fine powder was mixed with ~10 mg of $CaF_2$ (20 % by weight) and mounted on a low-level background holder according to the settings described in Supplementary material S1. Bivalves were then cut along the axis of maximum growth (from umbo to ventral margin), fixed in Styrene polymer and mounted on a glass slide with Epoteck® 301 epoxy glue



then trimmed and polished to a thickness of between 20-40 μm. Thin-sectioned bivalves thus prepared were used for petrographic observations, confocal Raman spectroscopy and laser ablation mass-spectrometry work. All the analyses described below targeted the ventral margin of the shells, the youngest material. The growth increments of *F. tenuicostata* are about twice those of *S. biradiata* at 1 mm and 0.5 mm respectively. Each measurement was made to encapsulate approximately the same growth period.

### 2.4 X-ray Diffraction Analysis

X-ray diffraction (XRD) analysis of powdered shell was performed using a PANalytical Empyrean Diffractometer operated at 40kV and 40mA (Cu Kα radiation and Ni Beta filter), using a PANalytical PixCel position sensitive detector. Particular settings are described in Supplementary material S1. Corundum NIST standard (676a, Certificate number 382200) was used as primary external standard and calcium fluoride ($CaF_2$) was used as secondary internal standard. The amount of secondary standard was tested by analysing different $CaF_2$: sample ratios. Since no major differences were observed between the different ratios tested, 20% weight of $CaF_2$ was chosen. $CaF_2$ (lattice parameter $a = b = c = 5.4662(2)$ Å, and $\alpha = \beta = \gamma = 90°$) is an ideal internal standard because its diffraction peaks do not interfere with those of $CaCO_3$.

Mineral phases were identified using the Inorganic Crystal Structure Database (ICSD) and International Centre for Diffraction Data (ICDD) database. The Rietveld refinement technique was used to quantify the phases. The Scherrer equation (Cullity, 1978) was used to calculate Coherently Scattering Domain (CSD) size (i.e. crystallite size) using the dominant reflections of calcite (104) and aragonite (111). Both the Rietveld refinement and the Scherrer calculation were calculated using PANanlytical HighScore Plus software (version 3.0e). The non-crystalline fraction, or amorphous material, in the powder was typically calculated from the overestimation of the secondary standard during Rietveld refinement (Scarlett and Madsen, 2006).

### 2.5 Confocal Raman Microscopy

Confocal Raman Microscopy (CRM) uses intense monochromatic light (here infrared laser) to irradiate samples. The subsequent molecular vibrations produce a highly specific Raman spectrum that reflects the molecular structure and the chemical identity of the samples. Changes in intensity and peak position indicate different vibrational modes of a molecule and therefor may reveal subtle differences in the crystalline form, especially between the different polymorphs of $CaCO_3$ (Nehrke and Nouet, 2011). $CaCO_3$ peaks measured using CRM are caused by the unit cell symmetry of the crystals and the molecular carbonate ions ($CO_3^{2-}$). Because of their shared $CO_3^{2-}$ related vibrational modes, differentiation between aragonite and calcite was accomplished using the lower end of the spectrum, 50 to 1200 cm$^{-1}$. In CRM a spectroscopy system is coupled with a confocal microscope allowing the mapping of samples with chemical sensitivity.

CRM was undertaken on thin sections of bivalve shells to confirm XRD-determined mineral phases and to provide key information about the crystallographic structure of $CaCO_3$. To mitigate high fluorescence levels caused by a green excitation laser (wavelength 532 nm) a diode infrared laser module (wavelength 785 nm) with 20x objective and 0.4





numerical aperture was used, coupled with the Raman imaging system (WITec Raman alpha300 RA+). CRM was performed in reflection mode whereby the scattered light is collected through a 100 μm detection fibre connected to a UHTS 300 spectrometer equipped with a 300 lines/mm grating. The full Raman spectra at each imaging pixel were acquired using a camera with a thermoelectrically cooled, back-illuminated CCD chip. The first order Raman peak of silicon (520.2 cm$^{-1}$) was

used to optimise alignment and signal intensity before each analysis. Measurements were made as large area scans with the accumulation of 3 to 5 spectra to minimise error ($\pm$ 1 cm$^{-1}$) and integration time of 0.1 to 0.050 s. Two spectral ranges were measured: 0 to 1600 cm$^{-1}$ and 1600 to 3000 cm$^{-1}$; the second range was explored in an attempt to detect organic functional groups (C-H or C=O) within the crystalline structures. CRM data were processed and analysed with the WITec ProjectFOUR software.

**2.6 Electron Probe Microanalysis**

Quantitative chemical maps were acquired on petrographic thin sections using a JEOL 8530F Hyperprobe equipped with five tuneable wavelength dispersive spectrometers. Operating conditions for instrument calibration comprise a 40 degrees take-off angle, beam energy of 15 kV and beam current of 20 nA. The beam was defocussed to 2 μm. Elements were acquired using analysing crystals: PET (pentaerythritol) for Ca K$\alpha$, S K$\alpha$, and Sr l$\alpha$, TAP (thallium acid phthalate) for Na K$\alpha$ and

Mg K$\alpha$. The standards used for instrument calibration were Barite for S K$\alpha$, Calcite for Ca K$\alpha$, Celestite for Sr l$\alpha$, Periclase for Mg K$\alpha$, and Jadeite for Na K$\alpha$. On peak counting times were 20 seconds and Mean Atomic Number background corrections were used throughout (Donovan and Tingle, 1996). Results are the average of 3 points and detection limits ranged from 0.008 wt % for S K$\alpha$ to 0.010 wt % for Mg K$\alpha$ to 0.049 wt % for Sr l$\alpha$. The Armstrong/Love Scott algorithm was used for data reduction (Armstrong, 1988). Quantitative element maps were obtained using the Probe Image® software

for X-ray intensity acquisition. The beam current was 80 nA with a 40 ms per pixel dwell time and a 4 x4 μm pixel dimension. Image processing and quantification was performed off-line with the CalcImage® software and output to Surfer®.

**2.7 Laser Ablation ICP-MS**

Analysis of carbonate material was undertaken using a Resonetics RESOlution M-50A-LR incorporating a Compex 102

excimer laser coupled with an Agilent 7700s quadrupole ICP-MS on shell material that had already been thin sectioned. Following two cleaning pulses and a 20 s period of background analysis, samples were spot ablated for 30 s at a 7Hz repetition rate, using a 75 μm beam and laser energy of 5 J cm$^{-2}$. Oxide polyatomic interferences were minimised by tuning flow rates for a ThO/Th of < 0.5%. The sample cell was flushed with ultrahigh purity He (350 mL min$^{-1}$) and N$_2$ (3.8 mL min$^{-1}$) and high purity Ar was employed as the plasma carrier gas. International glass standard NIST 612 and coral standard

MACS-3 were used as the primary reference materials, to calculate elemental concentrations (using stoichiometric $^{43}$Ca as the internal standard element) and to correct for instrument drift on all elements. Standard blocks were run after every 20





unknown sample. The mass spectra were reduced using Iolite (Armstrong, 1988). Data were collected on a total of 16 elements measured as $^{7}$Li, $^{24}$Mg, $^{28}$Si, $^{31}$P, $^{34}$S, $^{44}$Ca, $^{52}$Cr, $^{55}$Mn, $^{56}$Fe, $^{63}$Cu, $^{66}$Zn, $^{75}$As, $^{88}$Sr, $^{111}$Cd, $^{138}$Ba and $^{208}$Pb. Si concentration was used to determine if/when the laser hit the glass slide. The measurements taken when the laser hit the glass were discarded.

Trace element concentration profiles were measured in four shells of *F. tenuicostata* and two of *S. biradiata*. Five transects were measured on each shell: three parallel transects of seven to eights spots (T1–T3), oblique to the shell surface; one transect along the outer portion of the outer shell layer (T4, ten spots) and one along the inner portion of the outer shell surface (T5, ten spots; Fig. 2). Transects T1–T3 were measured to assess elemental variation towards the inner shell surface and the potential thickness of the last growth layer deposited. T4 and T5 were measured to compare trace element

concentrations in the innermost layer of the shell to the outermost layer.

## 3 Results

### 3.1 Shell Mineral Composition

XRD patterns indicate that the shells are composed of calcite and aragonite based on matching diffraction peaks (Fig. 3). The third identifiable diffraction pattern represents the internal standard, CaF$_2$ (20% weight of total sample). Rietveld refinement

was applied in order to obtain accurate phase quantification for each sample, however, the values produced by the software corresponding to the weight ratio of CaF$_2$ added were lower than the actual amount, and consequently, the results are not considered useful. Quantification errors in XRD are commonly caused by residual moisture, large organic content, amorphous crystallites or in this case errors due to small sample sizes and insufficient particle size reduction (Scarlett and Madsen, 2006). In the present case, moisture, organic material and amorphous crystals could all cause an over-estimation

(Scarlett and Madsen, 2006) but considered  the under-estimation was likely caused by the small sample size (~50mg). Accurate quantification on small sample sizes can be obtained using synchrotron technology. Although the phases cannot be quantified, the lattice parameters and crystallite sizes (i.e. coherently scattering domain, CSD) produced post-refinement and post-Scherrer calculation can be used because of the addition of the pre-calibrated CaF$_2$ standard.

All of the samples analysed were indexed to the orthorhombic symmetry system for aragonite and the hexagonal

system for calcite. The lattice parameters (*a, b, c*) generated by the Rietveld refinement differ from the geological reference materials for both calcite and aragonite. All samples of *F. tenuicostata* manifest expansion in aragonite and calcite along the *a*-axis (Fig. 4). All samples of *S. biradiata* but one (G170) show the same phenomenon. In contrast, G170 shows contraction along all axes. The *b*-axis in both phases and both species also generally show a positive distortion (stretching) with a few exceptions (*F. tenuicostata*: G159; *S. biradiata*: G170, G173, all three in aragonite only). The calcite *c*-axis shows shrinking

in both species. The distortion along the *c*-axis in aragonite is variable (two out of four *F. tenuicostata* show contraction and two out three of *S. biradiata*). The largest distortion was found in calcite, particularly for *a* and *b* (+0.12% to +0.02%); *c* only shows small distortions of -0.03% to -0.004%. Aragonite is distorted along the a-axis between +0.03% and +0.017%; *b*-



and *c*-axes manifest smaller distortions +0.014% and -0.008% maximum, respectively. The main difference between the two species considered here lies in the amount of distortion, whereby *F. tenuicostata* manifests smaller distortion compared to *S. biradiata* (Fig. 4). The maximum distortion in aragonite of 0.02% and 0.066% in calcite for *F. tenuicostata* (both *Δa/a*) and, 0.03% and ~0.13%, respectively in *S. biradiata* (also *Δa/a*). Furthermore, although minimal, *F. tenuicostata* shows

generalised stretching (0.004% maximum) along the *c*-axis in aragonite whereas *S. biradiata* shows shrinking (-0.008% maximum).

The CSD sizes estimated using the Sherrer equation (with K = 0.9) for the four dominant aragonite diffraction peaks corresponding to reflections (111), (102), (201) and (122) reveal aragonite crystals ranging from ~34 to ~144 nm long (Table 2). The majority of CSD values fall in the nano-crystal category (<100 nm) but some, especially those for the (122)

plane, belong to the larger ultra-fine category (100–500 nm).  The CSD sizes estimated for calcite used one dominant diffraction peak corresponding to reflection (104) to reveal smaller crystallites than in aragonite (~23 to ~29 nm long).

Different phases in the samples were identified using a combination of lattice modes and internal modes. Calcite and aragonite can be identified using the librational mode $L_c$ (282 cm$^{-1}$) and $L_a$ (209 cm$^{-1}$), respectively, and the in-plane band $\nu_4$ (713 cm$^{-1}$ for calcite and 702, 706 and 717 cm$^{-1}$ for aragonite); the shared peaks (translational mode, T, ~152 cm$^{-1}$ and

symmetry stretch, $\nu_1$, 1085 cm$^{-1}$) represent the $CO_3^{2-}$ ion motion and the C-O bond stretching common to both $CaCO_3$ polymorphs (Cuif et al., 2012, Fig. 5). Although very small and broad compared to aragonite-specific peaks, only the $L_C$ peak identified calcite. The most pronounced aragonite peak is $L_a$. The scan range of the spectrometer was increased to 3000 cm$^{-1}$ in order to detect the potential presence of organic compounds (e.g. Nehrke and Nouet, 2011), however, no features characteristic of these groups were found. As a consequence, the spectra presented here only extend to 2000 cm$^{-1}$.

Four filters were applied to each Raman spectrum (Fig. 5) to highlight different phases or reveal particular features of the shell: (1) Σ peak area of aragonite (centred at 209 cm$^{-1}$); (2) Σ peak area of calcite (centred at 282 cm$^{-1}$); (3) centre of mass/weighted width (Full Width at Half Maximum) of $\nu_1$ (centred at 1085 cm$^{-1}$) and (4) Σ peak area of the tail of the spectrum targeting fluorescence (centred at 1450 cm$^{-1}$). Peak intensities at 152 cm$^{-1}$ (translational mode T) and 209 cm$^{-1}$ were also ratioed (PIR) to distinguish crystal orientation variations (Fig. 5 and 6).

Aragonite and calcite were both found in all samples analysed using CRM. The small intensity of the calcite peak indicates that it is a minor phase. Two distinct types of spectra are visible for aragonite (Fig. 5), both with identical peak positions but different peak intensity ratios (PIR), (Fig. 5, 6F and 6F), between the translational mode T and the other peaks characteristic of aragonite (Fig. 5 and 6).

The full width half maximum (FWHM) of the symmetric stretch $\nu_1$ ($CaCO_3^{2-}$) reveals sequential variations or more

precisely alternating high/low peak width, especially in sample G154 (Fig. 6C and 6H). The other samples show more discrete changes in FWHM. Filter 4 (the filter showing fluorescence) reveals clear banding throughout samples (Fig. 6E and 6J).





### 3.2 Elemental Composition

EPMA results show total elemental compositions are consistently between 98 and 100 wt% (Fig. 6A, 6A). Calcium is generally consistent across the samples with an average concentration of ~39.5 wt% (Fig. 8B, 9B). S and Na comprise up to 0.16 and 0.66 wt% respectively, but importantly show variations in concentration with opposing trends, i.e. high Na

corresponds to low S and *vice versa* (Fig. 6). The concentration of S and Na correspond to growth lines with overall Na content higher towards the external surface of the shell and S towards the internal surface. Low S and high Na correspond to higher fluorescence levels detected using CRM (1450 cm$^{-1}$ filter, Fig. 6D and 6I). These results are consistent for both *S. biradiata* (Fig. 7) *and F. tenuicostata* (Fig. 8). Ca is typically uniformly distributed throughout each shell, nevertheless, patterns revealed by CRM correspond to patterns on Ca distribution maps. The patterns revealed by filters 4 and 5

(crystallinity and crystal orientation respectively) applied to the Raman spectrum, correspond to those visible on the Ca wt% map. Mg and Sr are below the instrumental detection limit (Table 3) and are not presented here.

Trace element concentration profiles measured using LA-ICP-MS showed NIST160 and MACS standard reproducibility in the range of acceptable values (between 0.5 and 4% on most elements; Supplementary materials S2). Very little reproducibility was observed between T1, T2 and T3 (Fig. 9 and 10) for all six shells tested however, compositionally

distinct layers were identified, as were differences between species (Fig. 11 and S3). In *F. tenuicostata*, ratios of Sr/Ca, S/Ca and Pb/Ca along transects T1–T3 generally increase from the external surface inwards (maximum +1 mmol/mol, +3 mmol/mol and +0.05 µmol/mol respectively; Supplementary materials S3). Only Sr/Ca in G158 decreases by 2 mmol/mol inwards (Appendix D-3). Mg/Ca, P/Ca and Ba/Ca (Fig. 11 and S3) follow the opposite trend, with a maximum decrease of 1.5 mmol/mol, 0.06 mmol/mol and 3 µmol/mol, respectively. The elemental ratios in *S. biradiata* all decrease along the T1–

T3 transects from the outer to the inner layer (Mg/Ca -0.15 mmol/mol, P/Ca -0.04 mmol/mol, Li/Ca -4 µmol/mol, Ba/Ca -2 µmol/mol) except Pb/Ca (+0.06 µmol/mol) and S/Ca, which is relatively stable at 1.3 mmol/mol. Sr/Ca increases inwards (+1.5 mmol/mol) but decrease to its original level at the edge of the internal layer (~2 mmol/mol). Li, Ca, Mn and Ba concentrations in *F. tenuicostata* are not linearly correlated to the other trace elements measured but statistically significant (paired t-test) linear correlation was found between [P]/[Sr] ($R^2$=0.70, p<2.2x10$^{-16}$), [P]/[S] ($R^2$=0.74, p<2.2x10$^{-16}$), [P]/[Mg]

($R^2$=0.55, p=1.283x10$^{-7}$) and [S]/[Mg] ($R^2$=0.55, p<2.2x10$^{-16}$). Li, Mg, Ca and Mn in *S. biradiata* are not linearly correlated to the other trace elements measured, but a statistically significant (paired t-test) linear correlation was found between [Sr]/[S] ($R^2$=0.74, p<2.2x10$^{-16}$), [Sr]/[Pb] ($R^2$=0.60, p<2.2x10$^{-16}$), [Sr]/[Ba] ($R^2$=0.92, p<2.2x10$^{-16}$), [Ba]/[S] ($R^2$=0.66, p<2.2x10$^{-16}$) and [Ba]/[Pb] ($R^2$=0.55, p<2.2x10$^{-16}$).

T4–T5 measurements confirm compositional differences between external and internal layers (Fig. 12 and S3).

Overall, the external layer of *S. biradiata* shells studied demonstrates higher ratios of Li/Ca, Mg/Ca, P/Ca, Sr/Ca and Ba/Ca compared to the internal layer. Only S/Ca and Pb/Ca do not follow this pattern, where S/Ca is the same in both layer and Pb/Ca is higher in the internal layer (Fig. 12 and S3). In comparison, *F. tenuicostata* shells show higher ratios of Mg/Ca,





Sr/Ca, S/Ca, Ba/Ca and Pb/Ca in their internal layer compared to their external layer. P/Ca decreases inwards and Li/Ca is essentially consistent.

S and P are indicators of organic content and their presence in the bivalves reflects a variation in macromolecules content throughout the shells. *F. tenuicostata* has a higher S-containing macromolecule content in the organic matrix of its

internal layer whereas a higher P/Ca ratio in the outer layer suggests organic matrix is richer in P-containing macromolecules in that section of the shell. Shells of *S. biradiata* are visibly more complex. They also show high S/Ca and low P/Ca towards the innermost layer but for most of the shell, variable S/Ca and P/Ca ratios suggest the external layers are characterised by alternating P- and S-containing macromolecules in organic components (Fig. 7–8 and 10–11).

## 4 Discussion

### 4.1 Mineralogy and Microstructure

Although *S. biradiata* and *F. tenuicostata* belong to subfamilies typically described as aragonitic (WoRMS et al., 2015), the XRD analysis and CRM results reveal a dominant aragonitic composition and a minor calcite fraction. The Rietveld refinement did not allow quantification of the different phases, however, CRM confirmed the presence of calcite by measuring (small intensities of the calcite librational mode $L_C$ compared to that of aragonite $L_a$). The Raman spectra of

*S. biradiata* have a slightly more intense calcite peak at 282 cm$^{-1}$ in comparison to *F. tenuicostata,* and the XRD pattern shows the main calcite peak in *F. tenuicostata* is generally sharper and offset towards 30°2θ compared to that of *S. biradiata*. According to Schroeder et al. (1969), this peak shift is indicative of the presence of Mg in carbonate, which suggests that some $Mg^{2+}$ ions have substituted for $Ca^{2+}$ in the $CaCO_3$ crystal lattice in *F. tenuicostata*. Considering the preferential incorporation of $Mg^{2+}$ into calcite as opposed to aragonite (Mucci and Morse, 1982), we conclude that although both species

appear to have similar amounts of calcite incorporated within their aragonitic shell, in *F. tenuicostata*, $Ca^{2+}$ ions have been partially substituted by $Mg^{2+}$ in the calcite lattice. This is not the case in *S. biradiata*. Sample G170 (*S. biradiata*), which shows shrinking in its aragonite phase according to XRD results, has a negative peak shift (CRM) indicative of $Sr^{2+}$ substitution in aragonite. The LA ICP-MS analysis of G170 was not undertaken because the thin section was too cracked towards the ventral margin, therefore, the higher Sr/Ca in G170 is assumed from the distortion measurements and CRM.

Aragonite is present in two different orientations in most of the *F. tenuicostata* samples. The combination of these two orientations (e.g. Fig. 6) clearly corresponds to the crossed-lamellar microstructure characteristic of many bivalve shells. The microstructure of *S. biradiata* shells consists of a prismatic outer shell layer, a crossed-lamellar middle layer and a complex crossed-lamellar inner layer (not present in the portion of the shell in Fig. 6). Both species precipitate their shells differently from a structural point of view. The precipitation of microstructural units in a crossed-lamellar pattern is

complex. The structural complexity of *F. tenuicostata* with its crossed-lamellar pattern associated with marked ribs suggests a higher level of biological control on $CaCO_3$ precipitation, which may influence the overall elemental composition of the shell. For example, Paquette and Reeder (1995), suggested that crystal surface structure has an effect on trace element





incorporation. As such, the different microstructures present in the two species studied here may be expected to yield different elemental compositions.

The measured lattice parameters reveal lattice distortion in both *F. tenuicostata* and *S. biradiata* compared to mineral $CaCO_3$. Pokroy et al. (2006) also reported lattice distortions (*Perna canaliculus*, *Acanthocardia tuberculata* and
*Strombus decorus*). After lattice parameter permutation, stretching was found along the *a*- and *b*-axes in both species, and shrinking was found along the *c*-axis in *S. biradiata* but not in *F. tenuicostata*. Distortion was not as pronounced in *S. biradiatata* and *F. tenuicostata* compared to the species studied by Pokroy *et al.* (2006) which can be explained by the use of annealing to relax the $CaCO_3$ structure. Maximum anisotropic distortion in aragonite is $\Delta a/a$ $1.1 \times 10^{-3}$, $\Delta b/b$ $2.5 \times 10^{-4}$ and $\Delta c/c$ $3 \times 10^{-4}$ for *F. tenuicostata*, and $\Delta a/a$ $1.7 \times 10^{-3}$, $\Delta b/b$ $7 \times 10^{-4}$ and $\Delta c/c$ $-6 \times 10^{-4}$ for *S. biradiata*. According to Pokroy *et al.*
(2006), the organic molecules could be the source of these structural distortions. However, given *F. tenuicostata*'s lattice is more stretched than that of *S. biradiata* the substitution of $Ca^{2+}$ ions by $Mg^{2+}$ ions described previously could also be a source of distortions. Ion substitution, which has been measured here using XRD, could also be a cause of this distortion because crystal structures accommodate trace elements by dilation or contraction of the ionic site (structural relaxation), which is a factor of flexibility and stability of the surrounding structure (Finch and Allison, 2007; Loste et al., 2003).

**4.2 Chemical Composition**

LA-ICP-MS results show that Mg is positively correlated to both S and P, and Sr to P in *F. tenuicostata*. These Mg correlations are not evident for *S. biradiata*, but Sr is positively correlated to S, Ba and Pb (Fig. 12), and Ba to S in this latter species.

The strong correlations found in the present study seem to indicate that Mg incorporation is influenced by both
phosphorus- and sulphur-containing macromolecules, i.e. both soluble and insoluble fractions of the organic shell matrix, in *F. tenuicostata* but not in *S. biradiata*. The incorporation of Sr appears to be influenced by the insoluble fraction of the organic matrix in *S. biradiata* and the soluble fraction in *F. tenuicostata*. The correlation found between Ba and S also suggests that the insoluble organic matrix influences the incorporation of Ba in *S. biradiata*. Shirai et al. (2014) suggested that incorporation of Sr is highly biologically mediated based on the strong correlation between Sr/Ca, S/Ca and shell
microstructure, and that elemental composition varies at a scale comparable to that of crystallites, i.e. the micrometre scale. The relatively coarse resolution used for LA-ICP-MS (75 μm) does not allow us to verify the spatial scale (<10 μm) at which the elemental composition of *F. tenuicostata* and *S. biradiata* varies but we can differentiate between the types of organic macromolecules influencing the shell geochemical composition, i.e. S-dominated or P-dominated. Our findings agree with that of Shirai et al. (2014) in that changes in microstructure and S/Ca ratio reflect changes in organic composition at the
mineralisation front. Our findings further show that P/Ca also reflects compositional changes at the mineralisation front. Consequently, we can say that not only do the organic matrices at the calcification front impact the chemical composition of the shell, but its composition also impacts the incorporation of trace elements in the shell, regardless of whether it is S- or P-dominated. This result is consistent with a number of previous studies that showed that S-containing and P-containing




macromolecules of the shell matrix, insoluble and soluble respectively, influence nucleation, growth and crystal structure (e.g. Borbas et al., 1991; Crenshaw, 1982; Reddy, 1977; Wheeler and Sikes, 1984). In the context of geochemical proxies, it becomes evident that the combined effect of shell organic matrix composition and environmental parameters (e.g. SST) may each cause variation in Mg/Ca and Sr/Ca ratios, two of the most common proxies used for reconstructing paleo-SST.

5        Additional complexity is also evident. P and S concentrations are consistent (transects T1, T2 and T3 comparison) in *F. tenuicostata* but highly variable in *S. biradiata*. In the latter, low Mg corresponds to high P and S, and high Sr to high S and low P. This suggests that the incorporation of these elements within the shell of *F. tenuicostata* is predominantly influenced by seawater, particularly Mg, whereas in *S. biradiata* it is predominantly influenced by the organic matrix composition, as demonstrated by the strong correlations in organic matrix-related elements The fact that *S. biradiata* is

visibly richer in organic matrix and owes its strength to its organic fraction whereas *F. tenuicostata* shells have a thicker and more complex structure (e.g. ridges), might be one explanation for this finding. Another remarkable point of comparison that might help understand the differences between the two species of interest lies in their taxonomy. Indeed, both species belong to the Heterodonta (subclass): Cardiida (order), but belong to different superfamilies: Cardioidea and Tellinoidea, *F. tenuicostata* and *S. biradiata* respectively (Ponder and Lindberg, 2008). Different organic content and shell organic matrix

composition might be explained by evolved differences in mineralisation processes (Cuif et al., 2011). However, without quantifying organic matrix components (total, soluble and insoluble) and identifying its composition, this remains speculative.

        These results not only reinforce the knowledge that the shell organic matrix of bivalves is involved in biomineralisation, but also that its level of influence depends on its composition (phosphorus-containing or sulfur-containing

macromolecules) and the specific element being incorporated into the $CaCO_3$. Furthermore, these results show this phenomenon is species-specific. The incorporation of the main elements (e.g. Mg, Sr, Ba) used as proxies from marine properties (e.g. SST) is highly mediated by the shell organic matrix in general.

## 5 Conclusions

The combination of geochemical, spectroscopic and crystallographic analyses on specimens of *F. tenuicostata* and *S.*

*biradiata* from King George Sound (south Western Australia) has revealed compositional and microstructural differences. Although the specimens were sampled from the same location and experienced the same marine conditions during their growth, they display different characteristics. Both species have been described as purely aragonitic but low levels of calcite and Mg-calcite are present in *S. biradiata* and *F. tenuicostata*, respectively. LA ICP-MS measurements revealed correlations between levels of Mg, S and P (*F. tenuicostata*), and Sr, S and Ba (*S. biradiata*). Although analysing the composition of the

shell organic matrix was beyond the scope of this study, the organic matrices of both bivalve species differ in composition and consequently in the way trace elements are incorporated into the $CaCO_3$. The incorporation of Sr and Ba is affected by the S-containing matrix of *S. biradiata* and the incorporation of Mg and Sr in *F. tenuicostata* is more influenced by S- and P-



containing organic macromolecules. The substitution of $Ca^{2+}$ by $Mg^{2+}$ in calcite in *F. tenuicostata* appears to be influenced by the composition of the organic matrix. The combination of S- and P-containing organics could be promoting ionic substitution in shells of *F. tenuicostata*.

The differences between these two species are clear and raise questions about genetic determinism considering they belong to different taxonomic superfamilies. This has consequences for environmental proxy applications. The present study shows that the elemental composition of marine bivalve shells is species-specific and influenced by multiple factors such as crystallographic structures, organic macromolecule composition and environmental setting. These factors complicate the use of bivalve shells as a proxy for reconstructing the past and present properties of seawater. Mg/Ca and Sr/Ca ratios are amongst the most common proxies used for SST reconstructions, however, this study shows elemental changes are not solely

caused by SST variation. High definition multimodal/multidisciplinary characterisation of both the organic and microstructure are central to understanding calcification and elemental incorporation. Only then can geochemical proxies be used to reconstruct true time-series and the "vital effect" accurately described.

Data availability

The data in this manuscript forms part of the PhD research of Liza Roger which is not yet submitted for examination. The thesis and all data generated as part of the thesis work will be available in the UWA Thesis Repository after examination.

Team list and contribution

L.M. Roger wrote the manuscript, collected the samples and analysed them. As PhD supervisors, A.D. George and J. Shaw

supervised the research and reviewed the manuscript. R. Hart helped understand and analyse the XRD data. M. Roberts collected the EPMA data. T. Becker helped understand the CRM data. B. McDonald and N. Evans collected the LA ICP-MS data.

Competing interests

All authors mentioned above declare having no competing financial, professional or personal interests that might influence the presentation of the work described in this manuscript.

Disclaimer

This manuscript is the result of the analyses that form part of the PhD research of L.M. Roger at the University of Western

Australia and the Centre for Microanalysis, Characterisation and Analysis in collaboration with the John de Laeter Centre of Curtin University, and is subject to interpretation.

Acknowledgements




The authors acknowledge the facilities, and the scientific and technical assistance of the Australian Microscopy & Microanalysis Research Facility at the Centre for Microscopy, Characterisation & Analysis, The University of Western Australia, a facility funded by the University, State and Commonwealth Governments, and the GeoHistory Facility, John de Laeter Centre at Curtin University, a collaborative research venture involving Curtin University, The University of Western Australia, Geological Survey of Western Australia and the Commonwealth Scientific and Industrial Research Organisation (CSIRO). The authors are also grateful for constructive comments and notes provided by B. Schöne, and the technical assistance provided by F. Nemeth. This study was financially supported by the School of Earth Sciences and the Centre for Microscopy, Characterisation and Analysis of the University of Western Australia.

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



**Table 1:** Lattice parameters *a,b* and *c* (Å) of aragonite after Rietveld refinement and other reference materials for comparison. *Perna canaliculus*: marine bivalve, *Acanthocardia tuberculata*: marine bivalve, *Strombus decorus persicus:* marine gastropod

| Origin | | *a* | *b* | *c* |
|---|---|---|---|---|
| *this study* | | | | |
| G156 | biogenic*: Fulvia tenuicostata* | 5.7471 | 4.9634 | 7.9673 |
| G158 | biogenic*: Fulvia tenuicostata* | 5.7481 | 4.9634 | 7.9678 |
| G159 | biogenic*: Fulvia tenuicostata* | 5.7484 | 4.9624 | 7.9712 |
| G160 | biogenic*: Fulvia tenuicostata* | 5.7480 | 4.9632 | 7.9692 |
| G170 | biogenic: *Soletellina biradiata* | 5.7413 | 4.9580 | 7.9545 |
| G171 | biogenic: *Soletellina biradiata* | 5.7519 | 4.9660 | 7.9629 |
| G173 | biogenic: *Soletellina biradiata* | 5.7514 | 4.9602 | 7.9686 |
| *Pokroy et al* * | | | | |
| ICSD-98-015-7993 | mineral: Morocco | 5.7420 | 4.9630 | 7.9680 |
| JCPDS-41-1475 | mineral | 5.7439 | 4.9623 | 7.9680 |
| ICSD-98-015-7994 | biogenic: *P. canaliculus* | 5.7520 | 4.9670 | 7.9640 |
| ICSD-98-015-7992 | biogenic*: A. tuberculata* | 5.7480 | 4.9650 | 7.9640 |
| ICSD-98-015-7995 | biogenic: *S. decorus persicus* | 5.7530 | 4.9690 | 7.9590 |
| ICSD-98-015-7996 | biogenic: *S. decorus persicus* (Bleached) | 5.7430 | 4.9630 | 7.9640 |
| ICSD-98-015-7997 | biogenic: *S. decorus persicus* (Annealed) | 5.7530 | 4.9690 | 7.9610 |
| *De Villiers (1971)*** | | | | |
| ICSD-98-001-5194 | mineral: Nevada | 5.7400 | 4.9670 | 7.9670 |
| *Bragg (1925)**** | | | | |
| ICSD-98-005-6090 | mineral: Hungary | 5.7300 | 4.9500 | 7.9550 |
| *External standard* NIST 676a Alumina | Synthetic powder | 4.7593 | 4.7593 | 12.992 |
| *Internal standard* CaF2 | Synthetic powder | 5.4664 | 5.4664 | 5.4664 |

* Pokroy B., Zolotoyabko E., Caspi E.N., Fitch A.N., von Dreele R.B. Chemistry of Materials (1, 1898-, 19, 3244 – 3251, (2007)

JCPDS: data file 41 – 1475, Joint Commettee on Powder Diffraction Standards – Keller, L., Rask, J., and Buseck, P., 1989, JCPDS card no. 41-1475, Arizona State Univ., Tempe, AZ, USA., ICDD Grant-in-Aid

** De Villiers, J.P.R. (1971), American mineralogist, 56, 758-766 – XRD from single crystal CaCO3 - Nevada

10 *** Bragg, W.L., Zeitschrift fuer Kristallographie, Kristallogeometrie, kristallphusik, kristallchemie (-144, 1977) 61, 425-451, (1925) – XRD from single crystal CaCO3 – Hungary



**Table 2**: Aragonite and calcite crystallite size (coherently scattering domain, CSD) in nm, estimated using the Sherrer equation (Cullity, 1978), with K=0.9, on 4 main diffraction peaks of aragonite and peak (104) of calcite

| Sample ID | CSD (nm) | | | | |
|---|---|---|---|---|---|
| | Aragonite peak (hkl) | | | | Calcite peak (hkl) |
| | (111) | (102) | (201) | (122) | (104) |
| *F. tenuicostata:* | | | | | |
| G155 | 68.5 | 58.8 | 41.6 | 105.2 | 25.5 |
| G156 | 64.2 | 69.9 | - | 58.3 | 26.2 |
| G158 | 69.1 | 64.4 | 42.3 | - | 28.3 |
| G159 | 45.6 | 50.1 | 35.6 | 34.4 | 29.1 |
| G160 | 66.3 | 73.0 | - | - | 26.2 |
| *S. biradiata:* | | | | | |
| G170 | 69.1 | 68.1 | 51.2 | - | 29.0 |
| G171 | 111.7 | 72.3 | 37.2 | 143.8 | 23.3 |
| G173 | 93.7 | 93.9 | 46.8 | 99.1 | 24.6 |





**Table 3**: EPMA detection limits (3σ) on bivalve samples. Mg and Sr contents in both *S. biradiata* and *F. tenuicostata* were below the limits of detection (see Fig. 8 and 9).

| Element (symbol) | *Fulvia tenuicostata* | *Soletellina biradiata* |
|---|---|---|
| Sodium (Na) | 0.1355 | 0.17 |
| Magnesium (Mg) | 0.102 | 0.127 |
| Sulfur (S) | 0.01 | 0.085 |
| Calcium (Ca) | 0.115 | 0.1425 |
| Strontium (Sr) | 0.47 | 0.595 |





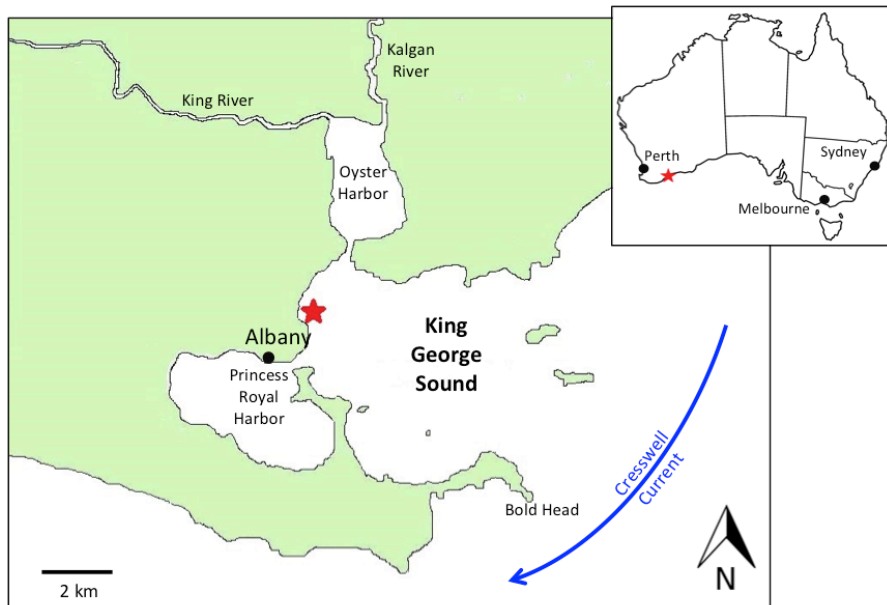

**Figure 1: Map of King George Sound and surrounding coastline, south Western Australia. The sampling site is indicated by the red star.**

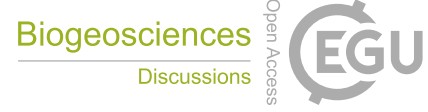

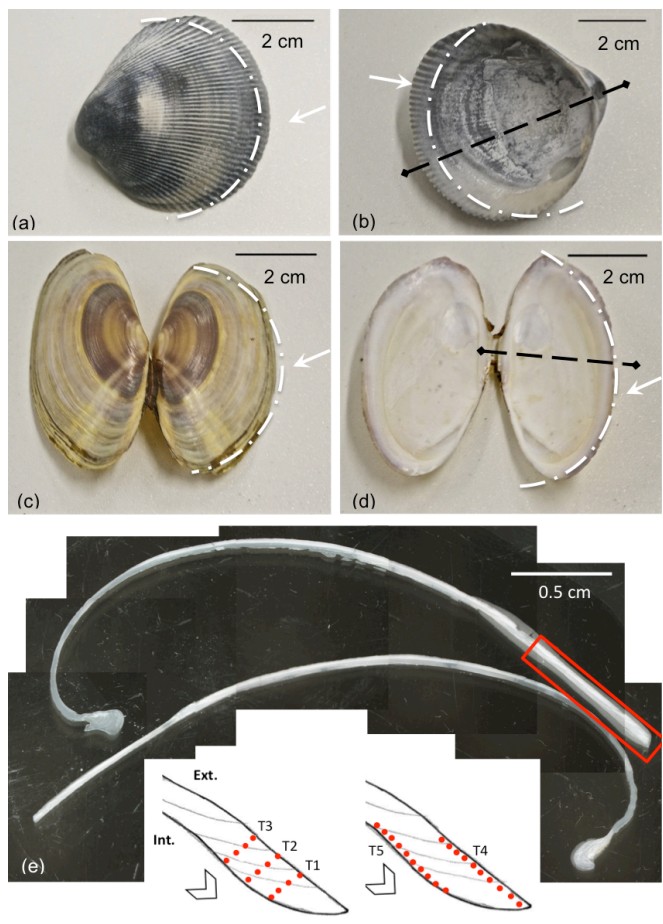

**Figure 2: Photographs of external (left) and internal (right) views of bivalves.** *Fulvia tenuicostata* (a,b); *Soletellina biradiata* (c, d). **(e) Photo-mosaic showing ventral margin (*S. biradiata*) as seen in thin section. Powder was sampled from the ventral margin for x-ray diffraction analysis (white dashed lines and arrows: a, b, c and d). The area adjacent to the ventral margin was then targeted for confocal Raman microscopy, LA-ICP-MS and EPMA (red box: e). Lower panel show representation of laser spots along five transects (T1–T5). T1–T3 are parallel to the shell edge and T4 and T5 are parallel to the external and internal shell margin (respectively).**





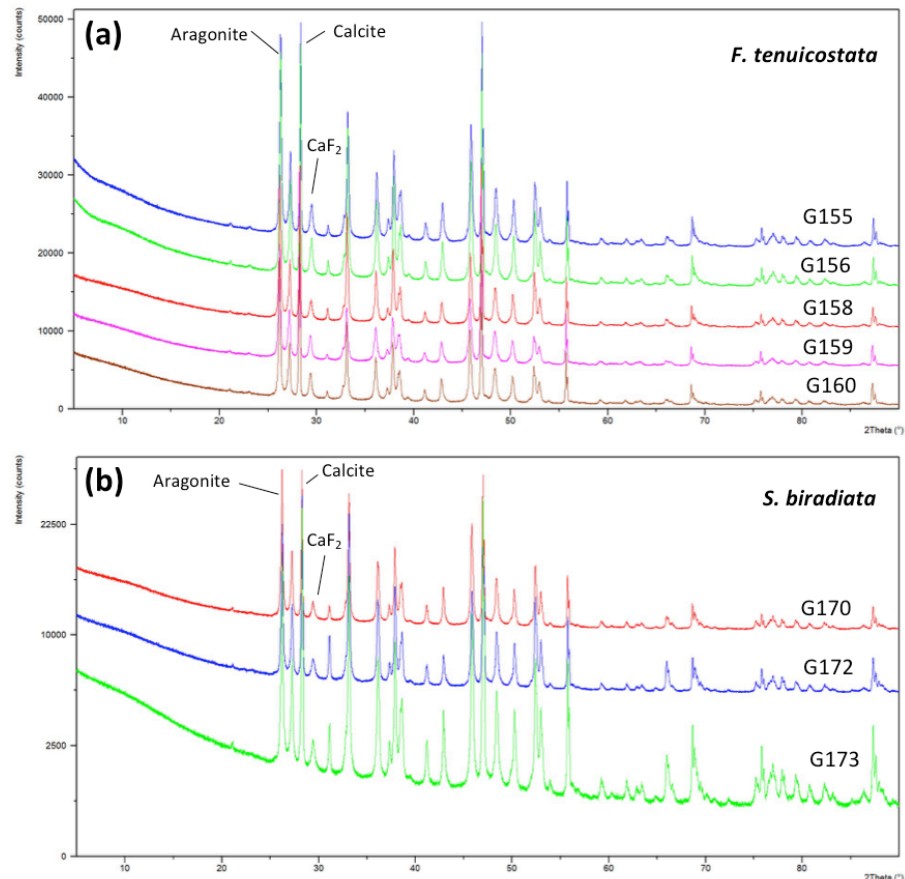

**Figure 3: X-ray diffraction patterns of specimens of *Fulvia tenuicostata* (a) and *Soletellina biradiata* (b). Each XRD pattern is offset from the next (along the y-axis) to show the consistency of the diffraction peak positions. Sample ID indicated to the right of each diffraction pattern. Intensity is relative. Note: G154, G157 and G172 were not analysed using XRD because the sample weight was too low. Main diffraction peaks indicated for each phase identified (aragonite, calcite and calcium fluorite $CaF_2$).**




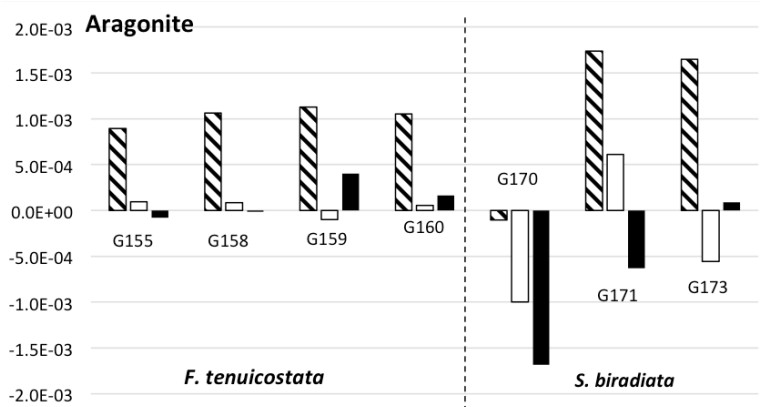

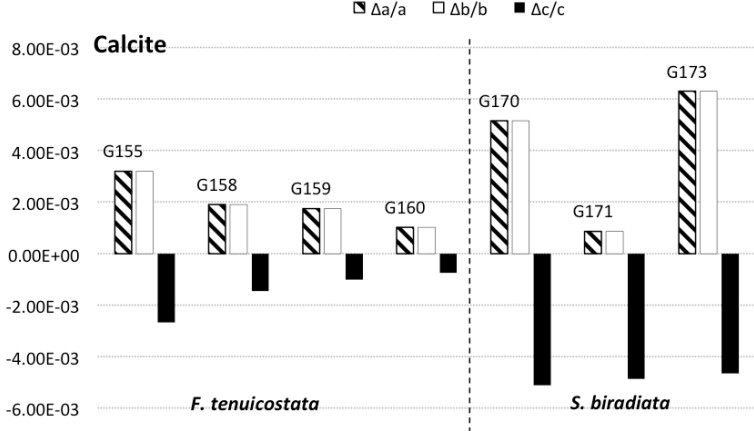

**Figure 4: Aragonite lattice distortion presented as Δ*a/a*, Δ*b/b* and Δ*c/c* compared to mineral standard (Pokroy et al, 2007) for four specimens of *F. tenuicostata* and three of *S. biradiata*. Y-axis indicates the amount of distortion compared to ICSD-98-015-7993 mentioned above. Negative distortions are indicative of shrinking, positive distortions are indicative of stretching.**





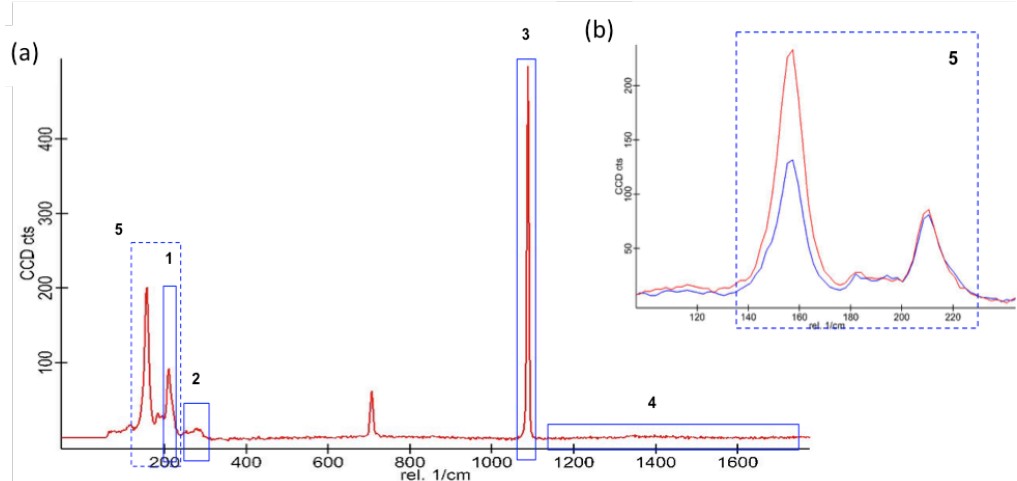

**Figure 5: Typical Raman spectra from *S. biradiata* and *F. tenuicostata* (a). Blue boxes correspond to filters applied to the data: filter 1 (Σ peak area) centred at 209 cm$^{-1}$ (width=36.6 cm$^{-1}$), filter 2 (Σ peak area) centred at 282 cm$^{-1}$ (width=66.3 cm$^{-1}$), filter 3 (centre of mass, weighted width) centred at 1085 cm$^{-1}$ (width=36.6 cm$^{-1}$), filter 4 (Σ peak area) centred at 1450 cm$^{-1}$ (width=600 cm$^{-1}$). The blue dashed box, filter 5, encapsulates filter 1 and a filter centred at 152 cm$^{-1}$ (Σ peak area, width=36.6 cm$^{-1}$), to represent their calculated peak intensity ratio. The blue and red spectra (b) show the peak intensity difference seen at 152 cm$^{-1}$ and 209 cm$^{-1}$. Each filter measures peak area, except filter 3, which measures Full Width at Half Maximum, and produces a qualitative map. Filter 1 targeted the aragonitic phase, filter 2 targeted calcite. Filter 3, by measuring variation in FWHM represents crystallinity and crystallite order. Filter 4, by targeting the tail of the spectrum shows fluorescence since peaks related to organics tend to show best in the tail of the spectrum. And finally, the peak intensity ratios (filter 5) show changes in crystallite orientation (aragonite crystallites)**




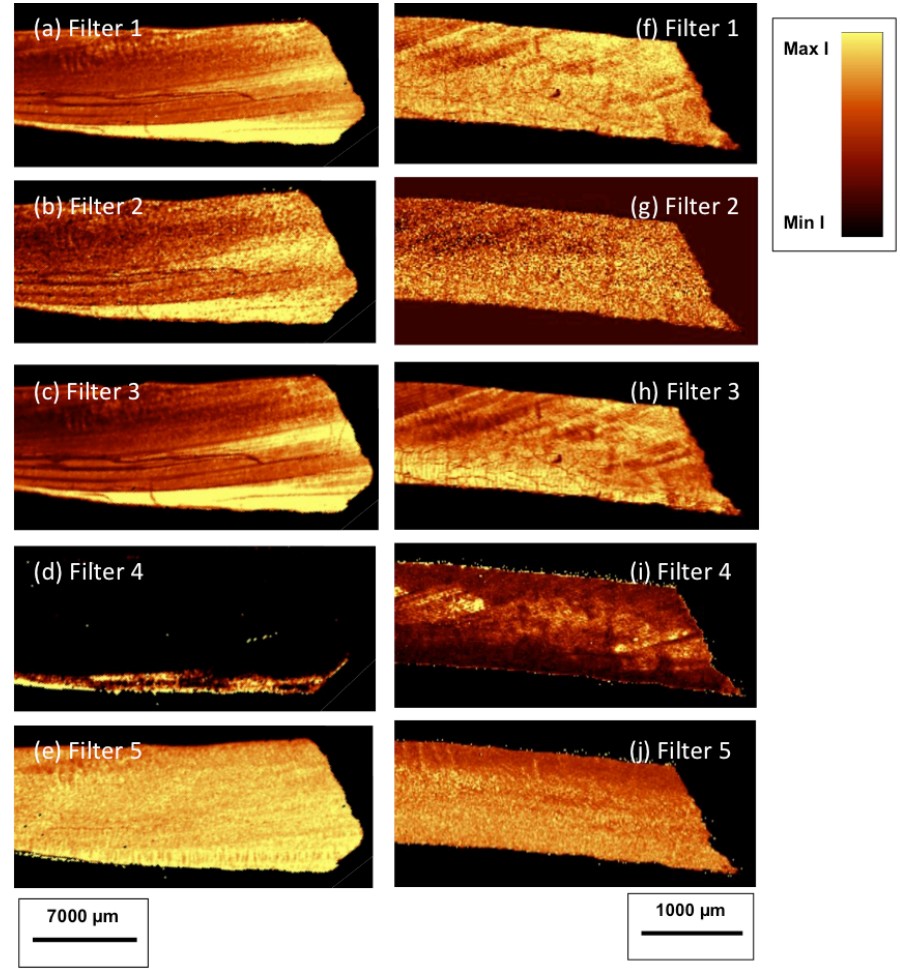

**Figure 6: Images extracted from filters applied to the Raman spectra (see Fig. 5) from samples G171 (*S. biradiata*) (a-e), and G155 (*F. tenuicostata*) (f-j). Scale bar: 7000 µm (a-e), 1000 µm (f-j). Colour scale: bright colours correspond to high intensities (high CCD counts) and dark colours correspond to low intensities (low CCD counts). Sub-horizontal and curved lines, most visible in (a)–(c) and (f)–(h), are structural features of the shells. Scale bar: 500 µm.**





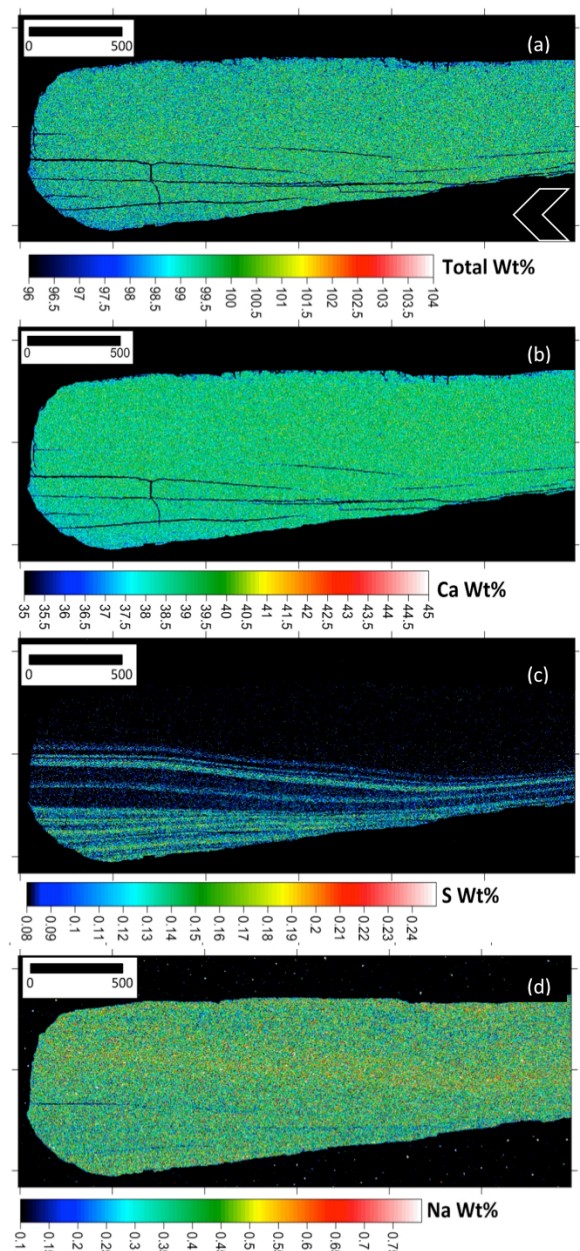





**Figure 7: EPMA elemental map of sample G171, *S. biradiata*: totals (a); Ca concentration (b); S concentration (c); Na concentration (d). Mg and Sr concentration was below instrumental detection limit therefor not present here. Arrow (a) indicated direction of growth. Note: this shell shows a thick edge compared to the adjacent shell section directly. *S. biradiata* shell thickness is variable along the cross-section.**





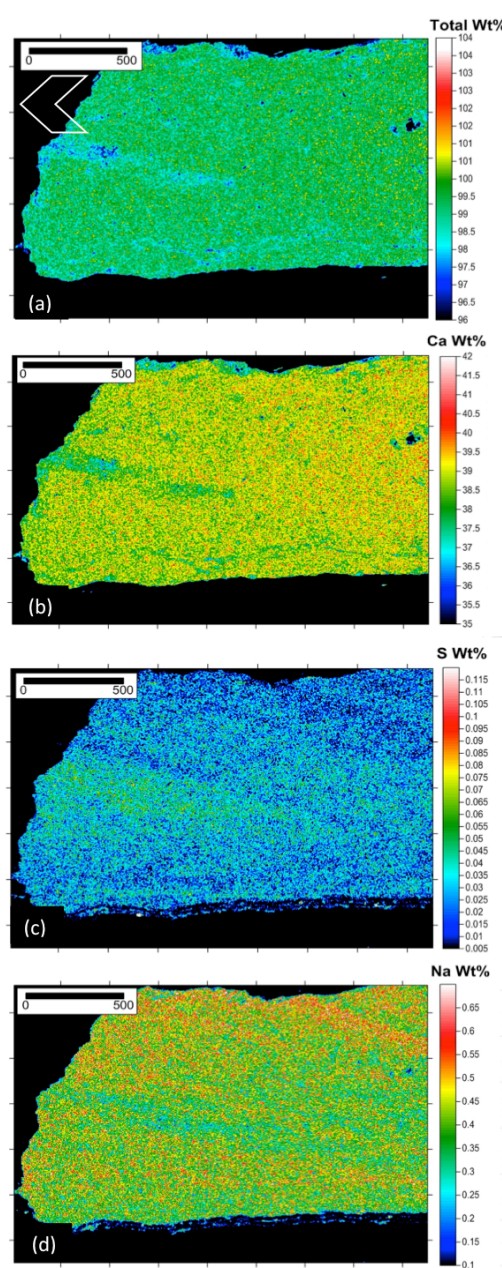



**Figure 8: EPMA elemental map of sample G155 (*F. tenuicostata*): totals (a); Ca concentration (b); S concentration (c); Na concentration (d). Mg and Sr were below instrumental detection limit therefor not presented. Dashed line indicates scan overlap, since the sample was scanned twice to optimize precisions and quality. Scale bar: 500 μm. Arrow indicates direction of growth.**



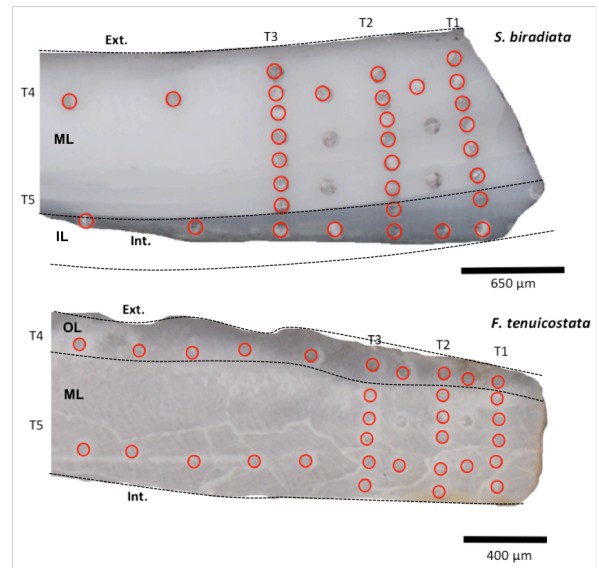

**Figure 9: Location of laser spots along five transects. T1–T3 transects towards the inner shell surface and parallel to each other. T4 along the outer portion of the outer shell layer and T5 along the inner portion of the outer shell layer. Inner and outer portion of the outer shell layer indicates by Int. and Ext. respectively. Shell layers indicated by OL: outer layer and ML: middle layer, and IL: internal layer. Note: absence of outer shell layer and inner shell layer in these sections of *S. biradiata* and *F. tenuicostata* respectively. T4 and T5 were sampled from different shell layers as labelled.**





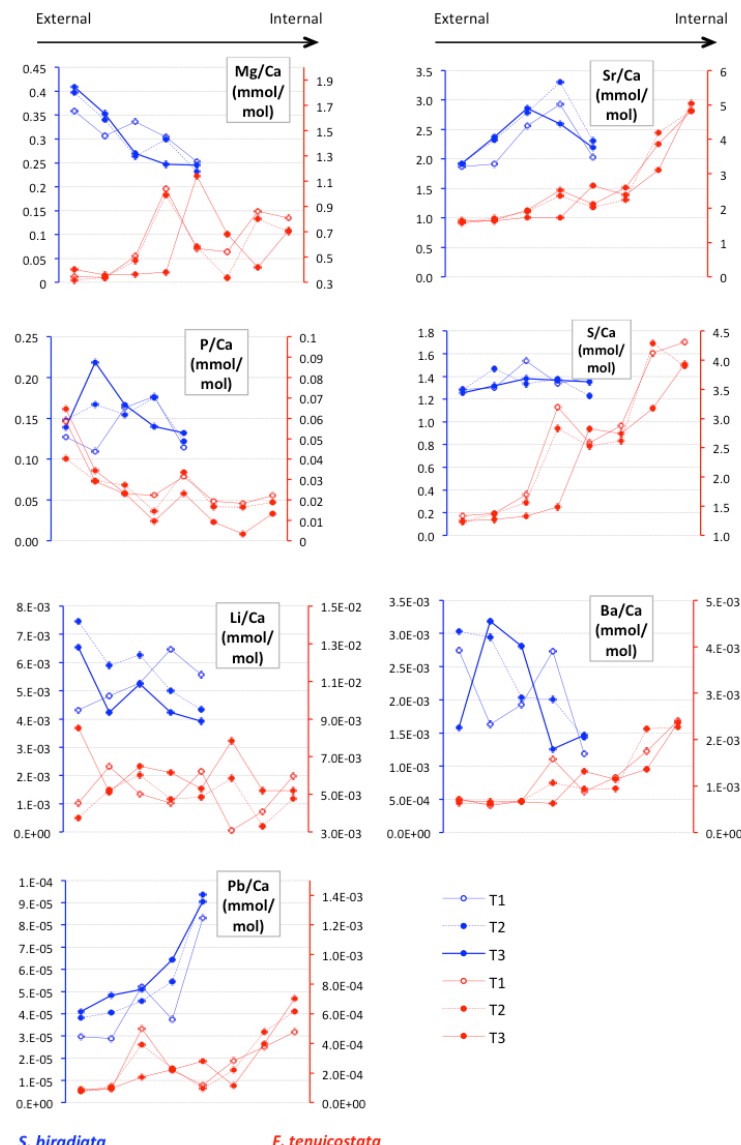

**Figure 10: LA-ICP-MS results for transect T1, T2 and T3. Blue lines represent results for *S. biradiata* (G171) and red lines represent results for *F. tenuicostata* (G155). Five laser spots/transect were done on *S.biradiata*.**



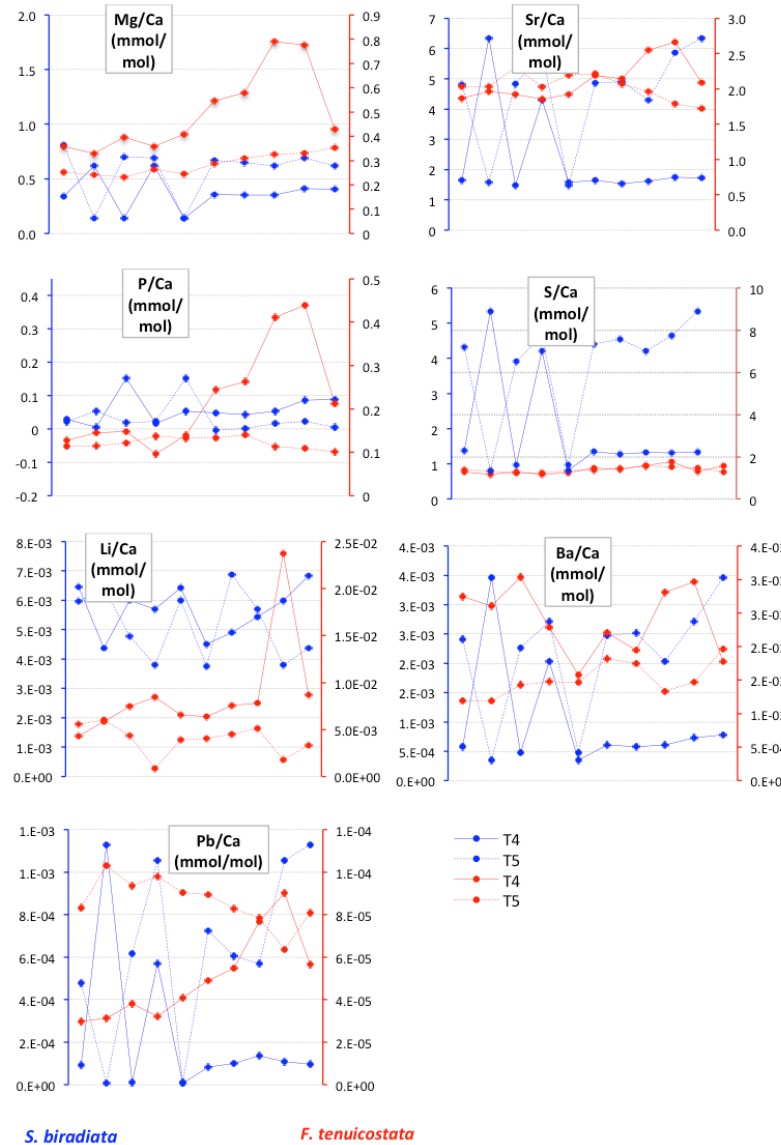

**Figure 11: LA-ICP-MS results for T4 and T5 transects. Blue lines represent results for *S. biradiata* (G171) and red lines represent results for *F. tenuicostata* (G155)**



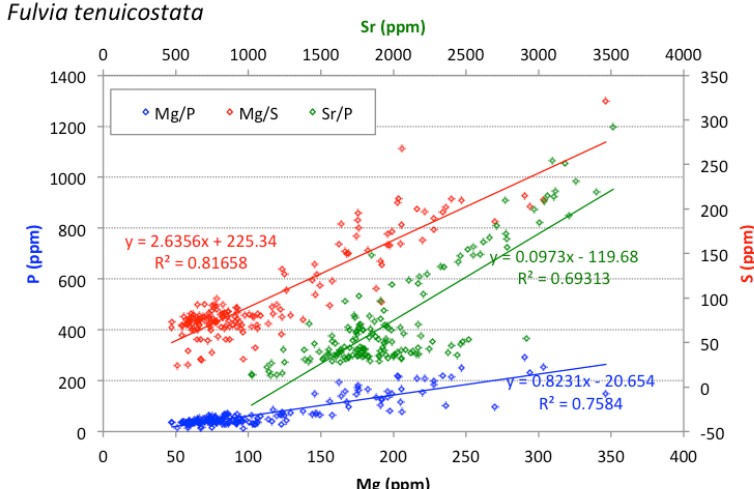

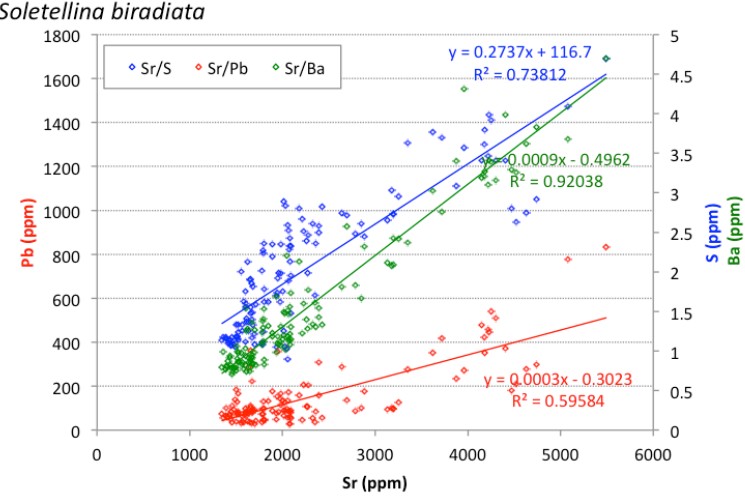

**Figure 12: Graphs showing significant linear correlations of LA-ICP-MS measurements for *F. tenuicostata* (Mg/P. Mg/S and Sr/P) and *S. biradiata* (Sr/S, Sr/Pb and Sr/Ba)**