# Peer review of "Geochemical and microstructural characterisation of two species of cool-water bivalves (*Fulvia tenuicostata* and *Soletellina biradiata*) from Western Australia."

_Biogeosciences, 2016_

## Referee Comment (RC1) · A. D. Wanamaker Jr. (Referee) · 8 Feb 2017

This is a very well prepared study that has been effectively communicated. In a general sense, the physical and chemical properties of microstructures in biocarbonates need to be better characterized in order for workers to best utilize these potential proxies. Recent advancements (including this study) in this area are facilitating a more comprehensive understanding of calcification processes including elemental uptake. Hence, this study is very appropriate for Biogeosciences and I think it will be of broad interest to many who work on biomineralization or those who use mollusk geochemical proxies

for their research.

The study is technically sound and warrants publication in BG.

Minor considerations:

p-values- perhaps it might be better to use a threshold of 95% or 99%, instead of such small values like 2.2 x10ˆ-16. Something like p < 0.01 or p< 0.001.

Figures 10/11- it is hard to see the uncertainty estimates in the elemental ratios. It might be useful to mention in the Figure caption text the uncertainty and that the uncertainty is generally smaller than the symbols.

Figure 12 could be improved by putting on 95% confidence intervals for each biplot. And make sure that the text is not covered by data (i.e., Sr/P and Mg/P in the top panel).

Line- 12- ocean quahogs (add Butler et al., 2013)

Butler, P.G., Wanamaker, A.D., Scourse, J.D., Richardson, C.A., Reynolds, D.J., 2013. Variability of marine climate on the North Icelandic Shelf in a 1357-year proxy archive based on growth increments in the bivalve Arctica islandica. Palaeogeogr Palaeocl 373, 141-151.

Line 15- the references for bivalve shell records used in environmental variability are a bit old- I might consider adding a few other references here from work in the last couple of years. For example:

Mette, M.J., Wanamaker, A.D., Carroll, M.L., Ambrose, W.G., Retelle, M.J., (2016) Linking large-scale climate variability with Arctica islandica shell growth and geo-chemistry in northern Norway, Limnology and Oceanography, 61(2), 784-764, doi:10.1002/lno.10252.

Reynolds, D.J., Scourse, J.D., Halloran, P.R., Nederbragt, A.J., Wanamaker, A.D., But-ler, P.G., Richardson, C.A., Heinemeier, J., Eiríksson, J., Knudsen, K.L., Hall, I.R.,

(2016) Annually resolved North Atlantic marine climate over the last millennium, Nature Communications, 7, 13502, doi:10.1038/ncomms13502.

Reynolds, D.R., Butler, P.G., Williams, S.M., Richardson, C.A., Scourse, J.D., Wanamaker, A.D., Jr., Austin, W.E.N., Cage, A.G., and Sayer, M., (2013), A multiproxy reconstruction of Hebridean (NW Scotland) spring sea surface temperatures between AD 1805 and 2010, Palaeogeography, Palaeclimatology, Palaeoecology,386, 275-285, doi:10.1016/j.palaeo.2013.05.029.

---

## Referee Comment (RC2) · Anonymous Referee #3 · 9 Feb 2017

I feel that the overall work of the manuscript by Roger, et al. entitled "Geochemical and microstructural characteristics of two species of cool-water bivalves (Fulvia tenuicostata and Soletellina biradiata) from Western Australia." is fair. Roger, et al. investigates the characteristics of shell mineralogy and geochemistry with micrometer spatial level on the two different marine specimens collected from the southern coast of Western Australia. To understand the relationship between shell microstructures and geochemical signals is highly important because with technical advancing of high resolution analysis on geochemical proxies, the scientists in related fields are realizing that the heterogeneities observed in the microstructures of biogenic carbonates are

not always influenced only by surrounding environmental changes. The authors show important solid results in this context, however, I am also feeing that this manuscript is lacked some important explanation and data for better understanding and contributing to the scientific communities especially of paleoclimates and paleoceanography. Therefore I believe that this works could be worth to be discussed among the scientists in related fields of sciences.

My suggestions to improve this work are bellow: 1. There is no detail environmental data demonstrated in this manuscript during the period of each shell growing including water temperature, salinity, and nutrients, and so on. The authors insist in introduction part that the geochemical signatures are so important to reconstruct the past changes in environment. However, it is not demonstrated and discussed well in this manuscript and makes difficult to judges whether any of difference and changes of geochemical composition in microstructures are not related to environmental changes or not. 2. There is no detail explanation of the localization of shell mineralogy. The finding different mineral phase other than aragonite but calcite and high magnesium calcite must be one of the most distinct results in this paper. The lack of this explanation leads difficult to understand the mechanisms of the formation of each of mineral phase. On the other word, if the authors could show the mineral phase could be varied with environmental changes, for example, it could be great finding in this wider area of science. 3. There is no direct evidence to explain the geochemical difference among the specimens and changes along growth direction. The organic materials in the shell microstructures seems to be one of reasons to explain it, but without any direct evidence, for instance, the content of organic materials, it is hard to conclude it. The demonstration of the geochemical results obtained by different technique such wet chemistry will help to explain this because laser ablation method is good to get high resolution data but difficult to avoid the material background effects including the content of organic materials.

---

## Author Comment (AC1) · 1 Mar 2017

Reviewer report 1

We thank Professor Wanamaker for his positive feedback and helpful suggestions. Each recommendation was carefully considered and made accordingly. Point-by-point modifications are described below.

1. p-values: perhaps it might be better to use a threshold of 95% or 99%, instead of such small values like 2.2 x10$^{-16}$. Something like $p < 0.01$ or $p < 0.001$

»The p-values presented in section 3.2. were changed to p<0.001.

2. Figures 10/11- it is hard to see the uncertainty estimates in the elemental ratios. It might be useful to mention in the Figure caption text the uncertainty and that the uncertainty is generally smaller than the symbols

»Uncertainty estimates were added in the caption below Fig.10 and 11.

3. Figure 12 could be improved by putting on 95% confidence intervals for each biplot. And make sure that the text is not covered by data (i.e., Sr/P and Mg/P in the top panel)

»Confidence intervals of 95% and prediction bands were added to the plots on Fig. 12 and the trendline equations were moved away from the data points to avoid overlap.

4. Line 12- ocean quahogs (add Butler et al., 2013)

»Butler et al 2013 was added on line 12 page 3.

5. Line 15- the references for bivalve shell records used in environmental variability are a bit old- I might consider adding a few other references here from work in the last couple of years. For example: Mette, M.J., Wanamaker, A.D., Carroll, M.L., Ambrose, W.G., Retelle, M.J., (2016) Linking large-scale climate variability with Arctica islandica shell growth and geochemistry in northern Norway, Limnology and Oceanography, 61(2), 784-764, doi:10.1002/lno.10252. Reynolds, D.J., Scourse, J.D., Halloran, P.R., Nederbragt, A.J., Wanamaker, A.D., But- ler, P.G., Richardson, C.A., Heinemeier, J., Eiríksson, J., Knudsen, K.L., Hall, I.R., C2 (2016) Annually resolved North Atlantic marine climate over the last millennium, Nature Communications, 7, 13502, doi:10.1038/ncomms13502. Reynolds, D.R., Butler, P.G., Williams, S.M., Richardson, C.A., Scourse, J.D., Wana- maker, A.D., Jr., Austin, W.E.N., Cage, A.G., and Sayer, M., (2013), A multiproxy reconstruction of Hebridean (NW Scotland) spring sea surface temperatures between AD 1805 and 2010, Palaeogeography, Palaeclimatology, Palaeoecology,386, 275-285, doi:10.1016/j.palaeo.2013.05.029.

»The reference suggested were added on line 15 page 3.

All changes visible in new text supplement

Please also note the supplement to this comment:
http://www.biogeosciences-discuss.net/bg-2016-343/bg-2016-343-AC1-
supplement.pdf

─────────────────────────────

---

## Author Comment (AC2) · 1 Mar 2017

Reviewer report 2

We thank reviewer No.2 for their constructive comments and recommendations. Each recommendation was carefully considered and changes made accordingly. Point-by-point modifications are described below.

1. There is no detail environmental data demonstrated in this manuscript during the period of each shell growing including water temperature, salinity, and nutrients, and so on. The authors insist in introduction part that the geochemical signatures are so

important to reconstruct the past changes in environment. However, it is not demonstrated and discussed well in this manuscript and makes difficult to judges whether any of difference and changes of geochemical composition in microstructures are not related to environmental changes or not.

»No detailed environmental data is presented here because this study uses comparison of shells (collected from live animals) that experienced the same seawater conditions throughout their lifespan. This study investigated similarities and differences within and between species' mineral and chemical composition on the basis that documentation of variations in modern shell composition is a pre-requisite of ancient material.

2. There is no detail explanation of the localization of shell mineralogy. The finding different mineral phase other than aragonite but calcite and high magnesium calcite must be one of the most distinct results in this paper. The lack of this explanation leads difficult to understand the mechanisms of the formation of each of mineral phase. On the other word, if the authors could show the mineral phase could be varied with environmental changes, for example, it could be great finding in this wider area of science.

»The detailed mineral composition was obtained using powder XRD which is a bulk method that does not provide spatial information. For crystallographic mapping CRM was used. Figure 6a shows a map of aragonite. Through peak intensity, Figure 6a shows where aragonite is concentrated and where it is not. The areas where the intensity is low are areas where calcite and Mg-calcite are present. The mapping of these phases is difficult considering the peaks overlap. The software used for the data analysis did not allow for the demixing of peaks. We agree that this would be a great finding but unfortunately, the software capabilities were too limited to show such findings.

3. There is no direct evidence to explain the geochemical difference among the specimens and changes along growth direction. The organic materials in the shell microstructures seems to be one of reasons to explain it, but without any direct evidence, for instance, the content of organic materials, it is hard to conclude it. The demonstration of the geo- chemical results obtained by different technique such wet chemistry will help to explain this because laser ablation method is good to get high resolution data but difficult to avoid the material background effects including the content of organic materials.

»We agree that there is no direct evidence and the organic matrix is one factor that can explain the microstructure found in the shells studied here. Wet chemistry is not particularly suited for molluscs because the organic matrix is composed of soluble and insoluble macromolecules. Some dissolve during wet chemistry and some do not. Also, we know from experience that insoluble macromolecules form a residue on the surface of the liquid when wet chemistry is used. The results found using this method still includes part of the soluble fraction of the organic matrix. The method that would help here would be a direct analysis of the composition of the organic matrix but this technique was too time costly to be included in the present work. It is definitely a consideration for future studies.

(All changes can be found in the supplement text)

Please also note the supplement to this comment:
http://www.biogeosciences-discuss.net/bg-2016-343/bg-2016-343-AC2-supplement.pdf

**Supplement:**

**Geochemical and microstructural characterisation of two species of cool-water bivalves (*Fulvia tenuicostata* and *Soletellina biradiata*) from Western Australia.**

Liza M. Roger[1,2], Annette D. George[1], Jeremy Shaw[2], Robert D. Hart[2], Malcolm Roberts[2], Thomas Becker[2,3], Bradley J. McDonald[4] and Noreen J. Evans[4].

[1]School of Earth Sciences, The University of Western Australia, Crawley, 6009, Australia
[2]Centre for Microscopy, Characterisation and Analysis, The University of Western Australia, Crawley, 6009, Australia
[3]Department of Chemistry, Nanochemistry Research Institute, Curtin University, GPO Box U1987, Perth, 6845, Australia
[4]Department of Applied Geology, John de Laeter Centre, TIGeR, Curtin University, Bentley, 6102, Australia

10 *Correspondence to*: Liza M. Roger (liza.roger@hotmail.fr)

**Abstract.** The shells of two marine bivalve species (*Fulvia tenuicostata* and S*oletellina biradiata*), endemic to south Western Australia, have been characterised using a combined crystallographic, spectroscopic and geochemical approach. Both species have been described previously as purely aragonitic, however, this study identified the presence of three phases, namely aragonite, calcite and Mg-calcite using XRD analysis. Data obtained via confocal Raman spectroscopy, electron probe microanalysis, and laser ablation inductively coupled plasma - mass spectrometry (LA ICP-MS) show correlations between Mg/S and Mg/P in *F. tenuicostata*, and Sr/S and S/Ba in *S. biradiata*. The composition of organic macromolecules that constitute the shell organic matrix (i.e. soluble phosphorus-dominated and/or insoluble sulphur-dominated fraction) influences the incorporation of Mg, Sr and Ba into the crystal lattice. Ionic substitution, particularly $Ca^{2+}$ by $Mg^{2+}$ in calcite in *F. tenuicostata*, appears to have been promoted by the combination of both S- and P-dominated organic macromolecules. The elemental composition of these two marine bivalve shells is species-specific and is influenced by many factors such as crystallographic structure, organic macromolecule composition and environmental setting. In order to reliably use bivalve shells as proxies for paleoenvironmental reconstructions, both the organic and inorganic crystalline material need to be characterised to account for all influencing factors and accurately describe the "vital effect".

Keywords

Geochemistry, microstructure, bivalve, organic macromolecule, crystal lattice, Mg/Ca, Sr/Ca, aragonite, calcite

Copyright statement

The content of this manuscript are protected by copyright. I hereby assign the copyright to the journal Biogeosciences on behalf of all authors.

[revised manuscript text omitted]

Reynolds, D. J., Butler, P. G., Williams, S. M., Scourse, J. D., Richardson, C. A., Wanamaker, A. D., Austin, W. E. N., Cage, A. G., and Sayer, M. D. J.: A multiproxy reconstruction of Hebridean (NW Scotland) spring sea surface temperatures

20 between AD 1805 and 2010, Palaeogeography, Palaeoclimatology, Palaeoecology, 386, 275-285, 2013.

Reynolds, D. J., Scourse, J. D., Halloran, P. R., Nederbragt, A. J., Wanamaker, A. D., Butler, P. G., Richardson, C. A., Heinemeier, J., Eríksson, J., Knudsen, K. L., and Hall, I. R.: Annually resolved North Atlantic marine climate over the last millennium, Nature communications, doi: 10.1038/ncomms13502, 2016. 2016.

[revised manuscript text omitted]
*.** **For visual clarity the uncertainty estimates are not presented on the graphics but are as follow: *S. biradiata* Li 1σ = 0.03, Mg 1σ = 0.37, P 1σ =0.15, S 1σ =2.1, Sr 1σ**

=1.6, Ba 1σ =1.28E-05, Pb 1σ = 7.78E-5, and *F. tenuicostata* Li 1σ = 0.05, Mg 1σ = 0.19, P 1σ = 0.25, S 1σ = 1.59, Sr 1σ = 1.20, Ba 1σ = 9.3E-04, Pb 1σ = 2.14E-04.

[Figure]

**Figure 11: LA-ICP-MS results for T4 and T5 transects. Blue lines represent results for *S. biradiata* (G171) and red lines represent results for *F. tenuicostata* (G155). For visual clarity the uncertainty estimates are not presented on the graphics but are as follow: *S. biradiata* Li 1σ = 0.03, Mg 1σ = 0.26, P 1σ =0.18, S 1σ =2.2, Sr 1σ =1.7, Ba 1σ =0.001, Pb 1σ = 2.24E-4, and *F. tenuicostata* Li 1σ = 0.05, Mg 1σ = 0.24, P 1σ = 0.3, S 1σ = 1.78, Sr 1σ = 1.0, Ba 1σ = 0.001, Pb 1σ = 1.0E-04.**

**Figure 12: Graphs showing significant linear correlations and 95% confidence intervals of LA-ICP-MS measurements for *F. tenuicostata* (Mg/P. Mg/S and Sr/P) and *S. biradiata* (Sr/S, Sr/Pb and Sr/Ba)**

---

## Author Response (AR1)

**Geochemical and microstructural characterisation of two species of cool-water bivalves (*Fulvia tenuicostata* and *Soletellina biradiata*) from Western Australia.**

Liza M. Roger[1,2], Annette D. George[1], Jeremy Shaw[2], Robert D. Hart[2], Malcolm Roberts[2], Thomas Becker[2,3], Bradley J. McDonald[4] and Noreen J. Evans[4].

[1]School of Earth Sciences, The University of Western Australia, Crawley, 6009, Australia
[2]Centre for Microscopy, Characterisation and Analysis, The University of Western Australia, Crawley, 6009, Australia
[3]Department of Chemistry, Nanochemistry Research Institute, Curtin University, GPO Box U1987, Perth, 6845, Australia
[4]Department of Applied Geology, John de Laeter Centre, TIGeR, Curtin University, Bentley, 6102, Australia

*Correspondence to*: Liza M. Roger (liza.roger@hotmail.fr)

**Abstract.** The shells of two marine bivalve species (*Fulvia tenuicostata* and *Soletellina biradiata*), endemic to south Western Australia, have been characterised using a combined crystallographic, spectroscopic and geochemical approach. Both species have been described previously as purely aragonitic, however, this study identified the presence of three phases, namely aragonite, calcite and Mg-calcite using XRD analysis. Data obtained via confocal Raman spectroscopy, electron

5   probe microanalysis, and laser ablation inductively coupled plasma - mass spectrometry (LA ICP-MS) show correlations between Mg/S and Mg/P in *F. tenuicostata*, and Sr/S and S/Ba in *S. biradiata*. The composition of organic macromolecules that constitute the shell organic matrix (i.e. soluble phosphorus-dominated and/or insoluble sulphur-dominated fraction) influences the incorporation of Mg, Sr and Ba into the crystal lattice. Ionic substitution, particularly $Ca^{2+}$ by $Mg^{2+}$ in calcite in *F. tenuicostata*, appears to have been promoted by the combination of both S- and P-dominated organic macromolecules.

10   The elemental composition of these two marine bivalve shells is species-specific and is influenced by many factors such as crystallographic structure, organic macromolecule composition and environmental setting. In order to reliably use bivalve shells as proxies for paleoenvironmental reconstructions, both the organic and inorganic crystalline material need to be characterised to account for all influencing factors and accurately describe the "vital effect".

15   Keywords

Geochemistry, microstructure, bivalve, organic macromolecule, crystal lattice, Mg/Ca, Sr/Ca, aragonite, calcite

Copyright statement

The content of this manuscript are protected by copyright. I hereby assign the copyright to the journal Biogeosciences on
20   behalf of all authors.

[revised manuscript text omitted]

**Reviewer reports**

**We are pleased that this manuscript has been accepted subject to minor revisions. We thank the editor and the reviewers for their positive feedback, comments and helpful suggestions. All comments and suggestions have been carefully considered and revisions made accordingly. Point-by-point modifications are described below.**

**Report from the Editor: the Editor suggested the addition of a statement about the microenvironment experienced by the bivalves to inform the reader further considering sediment environmental gradients can be steep and patchy. We added "from the muddy sands (600–800 µm grain size)" on line 20 of page 4. A reference to muddy sands can also be found at lines 4 and 9 of page 5.**

**No report from Reviewer 1**

**Report from Reviewer 2**

| Reviewer 2 Major Points |
|---|
| **1. I would like to request the authors to show shell layer distribution in Figure 9. Any molluscan shell consists of a few different layers, and each layer has a different microstructure with different elemental distribution. Normally two layers (outer and middle layers) are contained in a bivalve shell edge. So I guess transects T4 and T5 belong to different shell layers with different chemical nature (Bright and dark parts may represent different layers in Figure 9). Shirai et al. (2007), which is cited in this manuscript, presented a typical example.** |
| Reply: Reviewer 2 makes a good point regarding the shell layer distribution. The species presented here show similar shell structures similar to each other. To improve microstructural information, the following statements were added: page 5 line 7–9 "The shell structure of *F. tenuicostata* consists of a prismatic outer layer (OL), a simple crossed-lamellar middle layer (ML) and a complex crossed-lamellar inner layer (IL)" and line 12–15 "The shell structure of *S. biradiata* consists of a prismatic OL composed of acicular prismatic crystals directed toward the outer surface and locally curved, a simple crossed-lamellar ML and a complex crossed-lamellar IL. The OL may be absent, in which case the lamellae of the ML extend to the outer surface as noted for other species of this subfamily (Popov, 1986; Schneirder and Carter, 2001)". The light/dark pattern seen on *S. biradiata* of Fig. 9 corresponds to the difference in orientation and microstructure between the middle shell layers and the inner shell layer. Shell structure information was added to Figure 9 and in the caption. T4 and T5 were analysed along different layers and Figure 9 has been labeled to show this clearly. |
| **2. Shell microstructure of the two species should be mentioned briefly in Introduction or Results, if possible. Although the authors used the word "microstructure" in various parts of the manuscript, data on shell microstructure is not presented in this study. The term "shell microstructure" implies morphology of crystals and modes of their aggregation. For example, the most common types of molluscan shell microstructures include prismatic, nacreous, foliated and crossed-lamellar structures. Distinction of these microstructures seems important in the discussion of this paper, since organic contents are presumably different, depending on microstructures.** |
| Reply: The revisions of the text in Section 2.2 for point #1 have also addressed this comment. |
| **3. The authors mentioned that "the microstructure of *S. biradiata* shells consists of a nacreous outer portion of the** |

outer shell layer and a prismatic inner portion of the outer shell layer" (page 11, line 21). Probably the authors did not understand the shell microstructure of this species precisely. No species has been known to have a nacreous structure in the order Cardiida, and this description should be revised.

Reply: There is indeed a mistake in the description of the structure of *S. biradiata* which has now been corrected at the top of Section 4.1 (p. 11). This section now reads: "The microstructure of *S. biradiata* shells consists of a prismatic outer shell layer, a crossed-lamellar middle layer and a complex crossed-lamellar inner layer (not present in the portion of the shell in Fig. 6). Both species precipitate their shells differently from a structural point of view. The precipitation of microstructural units in a crossed-lamellar pattern is complex. The structural complexity of *F. tenuicostata* with its crossed-lamellar pattern associated with marked ribs suggests a higher level of biological control on $CaCO_3$ precipitation, which may influence the overall elemental composition of the shell. For example, Paquette and Reeder (1995), suggested that crystal surface structure has an effect on trace element incorporation. As such, the different microstructures present in the two species studied here may be expected to yield different elemental compositions."

**4. The following article should be cited regarding the shell microstructure of the Cardiidae (*Fulvia*).**
**Schneider, J. A. & J. G. Carter. 2001. Evolution and phylogenetic significance of Cardioidean shell microstructure (Mollusca, Bivalvia). Journal of Paleontology 75(3): 607-643.**

Reply: This publication has now been cited in the revisions for point #1.

**5. Source of data showing "purely aragonitic" (page 11, line 5): The first sentence of discussion started with "... are described as purely aragonitic (Boxshall, 2015)." I searched my library for literature but could not find any publication reporting shell mineral phase of *S. biradiata* and *F. tenuicostata*. The authors must quote the original data source for shell mineralogy of these two species. The reference cited in the manuscript (World Register of Marine Species) is a database of taxonomic names only, and there is no original data on shell mineralogy in the website.**

Reply: The text has been revised to read "Although *S. biradiata* and *F. tenuicostata* belong to subfamilies typically described as aragonitic (Schneirder and Carter, 2001)".

**6. As shown in Figure 2e, any bivalve shell becomes thinner towards its edge. From this viewpoint, the shell edge should be on the right side in Figure 7. Therefore, the growth direction indicated in this figure is opposite. If the left side of the figure is truly shell edge, it must be an artifact by sample treatment and needs explanation in the figure caption. Growth direction also should be checked in Figures 6, 7, 8 and 9 (especially Fig. 9 above).**

Reply: The authors thank reviewer for asking for clarifications concerning the growth direction in the figures. The shell presented in Fig. 7 (*S. biradiata*) has a thick edge like the other specimens of this species studied here. The layers of *S. biradiata* stack towards the edge of the shell; they do not thin as seen in most shells. The growth directions of the figures are, therefore, correct. A note was added to the figure caption to: "Note: this shell shows a thick edge compared to the adjacent shell section directly. *S. biradiata* shell thickness is variable along the cross-section."

**7. Laser spots: In Figure 2e, transects of laser spots T1 to T3 are obliquely aligned against the shell surface, but actually these transects are vertical to the shell surface in Figure 9. Figure 2e should be emended in this case.**

Reply: We agree that this is inconsistent and the T1 to T3 transects on Figure 2e have been amended to show their correct orientation matching Figure 9.

**8. LA-ICP-MS and Mg calcite: The following paper conducted elemental analyses of a molluscan shell with LA-ICP-MS, and their results can be compared with those of this study. In addition, there are many references reporting the presence of Mg calcite in molluscan shells.**
**Lazareth et al. 2007. Nyctemeral variations of magnesium intake in the calcite layer of a Chilean molluk shell (*Concholepas concholepas*, Gastropoda). Geochemica et Cosmochimica Acta 71: 5369-5383.**

| | |
|---|---|
| Reply: This publication has not been cited in the manuscript; it has not been used for Mg calcite because the study is based on gastropods rather than bivalves. | |
| **Reviewer 2 Minor points** | |

| Reviewer comments | Changes made |
|---|---|
| **Page 3, line 12: It is better to show scientific genus names together with common names for scientific publication: freshwater pearl mussels (*Margaritifera*), geoduck clams (*Panopea*), ocean quahogs (*Arctica islandica*), giant clams (*Tridacna*).** | Scientific genus names have been added. |
| **Page 3, line 20: isotopees > isotopes** | Revision made. |
| **Page 3, line 30: Schone > Schöne** | Revision made. |
| **Page 5, line 2: A common name must not be inserted between a species epithet and an author of a scientific name. *Fulvia tenucosta* ('thin-ribbed cockle' Lamarck, 1819) > *Fulvia tenuicostata* (Lamarck, 1819) ('thin-ribbed cockle'). An author name with parentheses means any change of generic status from the originally published name in zoological nomenclature. This is also true of *Soletellina biradiata*.** | *Fulvia tenuicostata* (Lamarck, 1819) ('thin-ribbed cockle') and *Soletellina biradiata* (Wood, 1815) ('double-rayed sunset clam', Fig. 2) |
| **Page 5, line 27: S1 > Supplementary material S1** | Revision made. |
| **References: Which should be used "." or "," between a title and a journal name? Two formats are mixed in the manuscript.** | Reference list punctuation has been changed to be consistent with journal style. |
| **Page 15, line 7: Tridacna > Italic** | *Tridacna* |
| **Page 15, Line 9: Growth Rate of Giant Clam Tridacna Gigas at Bikini Atoll as Revealed by Radioautography. > Growth rate of giant clam *Tridacna gigas* at Bikini Atoll as revealed by radioautography. (Tridacna gigas = Italic).** | *Tridacna gigas* |
| **Page 15, line 16: The citation of World Register of Marine Species is too long. According to WoRMS webpage, the recommended citation is WoRMS Editorial Board (2016). World Register of Marine Species. Available from http://www.marinespecies.org at VLIZ. Accessed 2016-XX-XX. doi:10.14284/170** | Revision made. |
| **Page 16, line 22: Pecten maximus > Italic** | *Pecten maximus* |
| **Page 16, line 26: Mytilus edulis > Italic** | *Mytilus edulis* |
| **Page 16, line 28: Saxidomus giganteus > Italic** | *Saxidomus giganteus* |

| | |
|---|---|
| **Page 16, line 30: Gillikin, 2005b, 2005c > These must be an identical paper.** | Gillikin, 2005c was deleted |
| **Page 16, line 49: Mytilus trossulus > Italic** | *Mytilus trossulus* |
| **Page 17, line 7: Isognomon ephippium > Italic** | *Isognomon ephippium* |
| **Page 17, line 17: Nerita undata > Italic** | *Nerita undata* |
| **Page 17, line 22: Journal of structural biology > Journal of Structural Biology** | Revision made. |
| **Page 17, line 28: Pinna nobilis > Italic** | *Pinna nobilis* |
| **Page 17, line 31: fo > of** | of |
| **Page 17, line 34: M. margarifitera > M. margaritifera (spelling correction) & Italic** | *M. margaritifera* |
| **Page 17, lines 36, 39, 43 & 45: Arctica islandica > Italic** | *Arctica islandica* |
| **Page 17, line 41: Schone > Schöne** | Schöne |
| **Page 18, line 1: Schone > Schöne** | Schöne |
| **Page 18, line 2: Arctica islandica > Italic** | *Arctica islandica* |
| **Page 18, line 16: Carbonate Calcification by Organic Matrix > carbonate calcification by organic matrix** | Revision made. |
| **Figure caption, Fig. 1: Star = locality?** | "The sampling site is indicated by the red star" added to figure caption (Figure 1) |
| **Figure caption. Fig. 4: ICSD-98-015-7993 appears twice in the caption. It is better to put "mineral standard" to the first one.** | The first "ICSD-98-015-7993" was replaced by "mineral standard" in the caption of Fig. 4 |
| **Figure 6: There is an explanation on arrows in the caption, but there is no arrow in the figure.** | Reference to arrows has been deleted. |
| **Figure 6a, b: There are a few sharp horizontal lines in the shell. Are these cracks artificially made or original structure? If they are artifacts, it is better to add explanation to the figure caption.** | The sharp lines are primary planes within the structure of the shells. "Sub-horizontal and curved lines, most visible in (a)–(c) and (f)–(h), are structural features of the shells. Scale bar: 500 μm." was added to the figure caption |
| **Figure 6: Add "Scale bar: 500 μm." to the figure caption.** | "Scale bar: 500 μm" has been added to the figure caption. |

**Report from Reviewer 3**

| |
|---|
| **Reviewer 3 Main Points** |
| **1. No detailed environmental data provided in this manuscript during shell growth including water temperature, salinity, and nutrients, and so on. It is not demonstrated in this manuscript and makes it difficult to judge whether difference and changes in microstructures are not related to environmental changes but other factors.** |
| Reply: This study uses comparison of shells (collected from live animals) that experienced the same seawater conditions |

throughout their lifespans. This study investigated similarities and differences within and between species' mineral and chemical composition on the basis that documentation of variations in modern shell composition is a pre-requisite for analysis of ancient materials.

**2. There is no detailed explanation of the localization of shell mineralogy. Yes, the finding different mineral phase other than aragonite but calcite and high magnesium calcite must be one of most distinct results in this paper. The lack of this explanation leads difficult to understand the mechanisms of mineral phase.**

Reply: The detailed mineral composition was obtained using powder XRD which is a bulk method that does not provide spatial information. For crystallographic mapping CRM was used. Figure 6a shows a map of aragonite. Through peak intensity, Figure 6a shows where aragonite is concentrated and where it is not. The areas where the intensity is low are areas where calcite and Mg-calcite are present. The mapping of these phases is difficult considering the peaks overlap. The software used for the data analysis did not allow for the demixing of peaks. We agree that this would be a great finding but unfortunately, the software capabilities were too limited to show such findings.

**3. There is no direct evidence to explain the geochemical difference among the specimens and changes along growth direction. The organic materials in the shell microstructures seem to be one of reasons to explain it, but without any direct evidence, for instance, the content of organic materials, it is hard to conclude it. The demonstration of the geochemical results obtained by different technique of wet chemistry will help to explain this because Laser ablation method is good to get high resolution data but difficult to avoid the material background effects including the content of organic materials.**

Reply: We agree that there is no direct evidence and the organic matrix is one factor that can explain the microstructure found in the shells studied here. Wet chemistry is not particularly suited for molluscs because the organic matrix is composed of soluble and insoluble macromolecules. Some dissolve during wet chemistry and some do not. Also, we know from experience that insoluble macromolecules form a residue on the surface of the liquid when wet chemistry is used. The results found using this method still includes part of the soluble fraction of the organic matrix. The method that would help here would be a direct analysis of the composition of the organic matrix but this technique was too time costly considering the time frame associated to this project.